# Coffee Beverage: A New Strategy for the Synthesis of Polymethacrylates via ATRP

**DOI:** 10.3390/molecules27030840

**Published:** 2022-01-27

**Authors:** Karolina Surmacz, Paweł Błoniarz, Paweł Chmielarz

**Affiliations:** 1Doctoral School of Engineering and Technical Sciences, Rzeszow University of Technology, Al. Powstańców Warszawy 8, 35-959 Rzeszow, Poland; d503@stud.prz.edu.pl; 2Department of Physical Chemistry, Faculty of Chemistry, Rzeszow University of Technology, Al. Powstańców Warszawy 6, 35-959 Rzeszow, Poland; bloniarz@prz.edu.pl

**Keywords:** caffeine, DMAEMA, dually controlled reducing mechanism, ARGET ATRP, aqueous medium

## Abstract

Coffee, the most popular beverage in the 21st century society, was tested as a reaction environment for activators regenerated by electron transfer atom transfer radical polymerization (ARGET ATRP) without an additional reducing agent. Two blends were selected: pure Arabica beans and a proportional blend of Arabica and Robusta beans. The use of the solution received from the mixture with Robusta obtained a high molecular weight polymer product in a short time while maintaining a controlled structure of the synthesized product. Various monomers with hydrophilic characteristics, i.e., 2-(dimethylamino)ethyl methacrylate (DMAEMA), oligo(ethylene glycol) methyl ether methacrylate (OEGMA_500_), and glycidyl methacrylate (GMA), were polymerized. The proposed concept was carried out at different concentrations of coffee grounds, followed by the determination of the molar concentration of caffeine in applied beverages using DPV and HPLC techniques.

## 1. Introduction

Due to the still increasing pace of life and the need for stimulation of the body with caffeine, coffee has become an attractive drink among the society of the 21st century. In addition to its energy values, coffee is also appreciated for its taste and high content of antioxidants, which play a beneficial role in the human body. Coffee beans are a rich source of biologically active compounds, among which, apart from caffeine, one can mention chlorogenic acids, nicotinic acid, cafestol, kahweol, and trigonelline [1], which have significant antioxidant activity. The type of grain used and the method of its processing affect the antioxidant potential of a coffee brew. The highest number of compounds with antioxidant potential was determined in medium roast coffee grains due to the balance between degradation and the formation of Maillard’s reaction products, which are strong antioxidants [1,2]. There are over five hundred species of coffee trees known in the world, but only two of them are the most popular: coffee arabica (Arabica) and coffee canephora (Robusta). The Arabica variety, originating from Ethiopia, is appreciated by baristas for its delicate flavor, which results from its relatively low caffeine content. Coffee canephora comes from central and western sub-Saharan Africa [3], has a rather sharp and distinctive aroma because its beans are richer in caffeine and phenolic compounds [1,4]. However, it should also be noted that byproducts of coffee processing, i.e., coffee silverskin and spent coffee extract contain a large number of antioxidants [1,5], the management of which is one of the tasks of green chemistry. When looking for techniques to use readily available antioxidants in order to synthesize macromolecular compounds, it is worth using the atom transfer radical polymerization with activators regenerated by electron transfer (ARGET ATRP) [6,7,8,9,10]. This technique makes it possible to obtain polymers with a predetermined structure and properties in a controlled manner with a simultaneous low concentration of the catalytic complex both in organic solvents [11,12] and in environmentally friendly aqueous systems [13,14,15,16,17,18]. The reactions carried out in ionic liquids were reported as well [19]. The reducing component in ARGET ATRP is a compound with redox properties enabling easy regeneration of the catalytic complex [17]. So far, the most-used are vitamins like ascorbic acid [20,21,22,23] or riboflavin [12,24] and sugars, such as glucose [25] and cyclodextrin [26], but also the novel external reducing agents, such as liquid metal [27].

Searching for a suitable reaction medium is no less important for sustainable development and proper waste management [15,28,29]. Depending on the selected method, ATRP can be carried out in a solution or emulsion medium. Equally important is the solubility of the monomers in the individual media, which determines the selection of the reaction solvent. Recently, ATRP of 2-(dimethylamino)ethyl methacrylate (DMAEMA) were successfully carried out in alcoholic beverages [30], ethanol/water [30] and acetone/water mixture [31], as well as in environmentally harmful solvents such as *N,N*-dimethylformamide (DMF) [32,33], dimethyl sulfoxide (DMSO) [34], toluene [35], anisole [36] or methanol [37]. Despite the polymerization carried out for hydrophilic monomers, the application of pure water as a polymerization medium is still difficult. It results from the occurrence of competitive reactions as termination of propagating radicals, hydrolysis of the alkyl halide, or disproportionation of Cu^I^ [38,39].

In this study, the coffee solution containing additional antioxidants was successfully used as an electron donor in the ARGET ATRP technique for the synthesis of high-molecular compounds of hydrophilic nature. Extract of two blends, i.e., pure Arabica beans and a proportional blend of Arabica and Robusta beans were used as a reaction medium, followed by the determination of the molar concentration of caffeine in applied beverages using DPV and HPLC techniques. Due to the pro-ecological and economical approach, the presented system of hydrophilic monomers polymerization has application potential, especially in the peri-biological field.

## 2. Results

The mechanism of a coffee-induced ATRP based on radicals generation process involving a copper−amine complex (activator) in its lower oxidation state, Cu^I^/L, is illustrated in Figure 1. The catalytic complex activates ethyl 2-bromoisobutyrate (EBiB) as an alkyl halide initiator or dormant chain end (P_n_−X), forming propagating radicals and oxidizing to a higher oxidation state.

Antioxidants derived from a coffee extract such as caffeine, chlorogenic acids, cafestol, kahweol, and trigonelline trigger a reduction of a deactivator complex, X-Cu^II^/L, ensuring continuous catalytic reaction. Voltammetric measurements were carried out, which confirmed the strong antioxidant effect of caffeine in both mixtures, therefore the mechanism in which caffeine is the dominant antioxidant factor controlling the polymerization process was presented. Caffeine, being the main antioxidant present in coffee, is oxidized in the reaction environment with the simultaneous incorporation of oxygen derived from the reaction solvent. The process takes place in a mild oxidizing medium of copper(II) bromide, where, as in the permanganate environment, a two-electron redox process takes place for the caffeine molecule. The mechanism involves the interaction of the oxidant—deactivator with the caffeine molecule to give a free radical derived from caffeine. Then, water particles from the reaction solvent react with a free radical, giving 8-hydroxy caffeine which fast tautomerise to the final oxidation product of caffeine—1,3,7-trimethyluric acid (Figure 1) [40]. According to the ATRP mechanism, the dynamic equilibrium of the copper(I) and copper(II) complex is involved in controlling of the polymerization. The complex of copper(II) introduced into the polymerization system undergoes reduction due to the caffeine action. Additionally, trace amounts of oxygen could cause oxidation of the copper and affect the dynamic equilibrium of Cu^II^ to Cu^I^ estabilishment. The polymerization process is then controlled by an appropriate ratio of activator to deactivator species [8,23].

Interestingly, the tertiary amine groups of DMAEMA also act as an internal reducing agent according to the ARGET ATRP mechanism [30]. The deactivator complex oxidizes the tertiary amine moiety (RN(CH_3_)_2_) of DMAEMA into radicals (R_1_R_2_N-ĊH_2_), which initiate the polymerization and then reduce to an activator complex [31]. As presented in the publication [31], the oxidation reaction of tertiary amines, catalyzed with copper ions (Cu^2+^), can take place under the reaction conditions. This transformation converts the inert C_α_-H-position of amines to an active radical position via in situ oxidative dehydrogenation and forms a polymerization initiation site, caused by chain branching and an increase in molecular weight distribution of the PDMAEMA chain. The presence of two types of internal control compounds is indicative of a dually controlled mechanism of ARGET ATRP.

### 2.1. ARGET ATRP of DMAEMA in Various Coffee Bean Blend Solutions

ARGET ATRP of DMAEMA was carried out in 10 °C and in room temperature (Appendix A). As expected, polymerization reaction in 10 °C was slower and gives higher molecular weight distribution of the obtained polymer (compare *k*_p_^app^ and *Ð* in Appendix A). Therefore, all next polymerization processes were performed at room temperature, achieving products in controlled manner and in a shorter time.

In an efficient ATRP catalyst system, the metal center must have two easily accessible oxidation states and must be able to accept the halogen, which requires a sufficiently high halogenophilicity [41]. Low halidophilicity, i.e., the association of the halide anion to the catalyst in the higher oxidation state, related to the connection of the halogen ion create a possibility of disproportionation of Cu^I^/L to X-Cu^II^/L and Cu^0^ [42,43]. Finally, halide anions must have a high affinity for Cu^II^L^2+^, such as to provide the presence of enough concentration of deactivator to give a well-controlled process. In aqueous systems, the X-Cu^II/^L bond can easily dissociate to Cu^II^L^2+^ and Br^−^, and typically required is a high concentration of catalyst or the presence of salts with halide anions in order to shift the equilibrium toward the X-Cu^II^/L deactivator species [28,44]. Because the experiments were carried out with a low ppm concentration of the catalyst (9.5 ppm by wt), the effect of the addition of a halide salt on the polymerization process in the medium of a coffee blend extract was investigated (Arabica & Robusta 1:1). Indeed, the lack of a halide salt caused a loss of process control due to ineffective catalyst regeneration, the addition of NaBr resulted in an increase of control over polymerization reflected by a valid initiation of efficiency values and a narrower molecular weight distribution (Appendix A). Higher salt concentration created more effective catalyst regeneration and increased control over the process (compare initiation efficiency in Appendix A); however, the reaction stopped after 0.5 h and did not reach a satisfactory MWD (Appendix A), therefore it was decided to conduct subsequent reactions at a sodium bromide concentration equal to 0.1 M.

For the synthesis in pure water without the addition of a reducing agent (Table 1, entry 1) a linear increase of conversion with time was observed (Figure 2). This is due to the reducing properties of DMAEMA, in which a tertiary amine group acts as an intrinsic reducing agent according to the ARGET ATRP mechanism. On the other hand, the N-CH_2_ group remains the initiation site of radical polymerization, resulting in a loss of ATRP control as indicated by an almost 900% initiation efficiency. Polymerization of tertiary amine monomers in coffee extract implies the dually controlled mechanism of ARGET ATRP through the action of using caffeine and monomers as a reducing agent. This dually controlled polymerization system (Table 1, entries 2–3) reduces initiation efficiency by more than three times, which is related to the higher control of polymerization while simultaneously obtaining higher monomer conversion and polymerization rate (compare *k*_p_^app^ in Table 1).

In the search for a suitable coffee mixture for ATRP, commercially available coffee blends were chosen in order to prepare the extracts for PDMAEMA synthesis (Table 1). Polymerizations were carried out in solutions obtained from two different mixtures of coffee grain from the Segafredo Zanetti corporation. It was purchased from Segafredo Espresso CASA as a blend of Arabica and Robusta grain (1:1) (Table 1, entry 2) and Segafredo Arabica (Table 1, entry 3) as a pure Arabica grains mixture. For both tested mixtures, the plot of the natural logarithm of the monomer concentration versus polymerization time was close to linear, proving a first-order kinetic model (Figure 2a). In both cases, despite the weaker control of the polymerization process (*Ð* = 1.63 and 1.88) caused by a low amount of catalytic complex, 9.5 ppm by wt, obtained PDMAEMA with monomodal weight distribution. Interestingly, the use of Arabica and Robusta blends obtains significantly longer polymer chains (*M*_n_ = 62,800) with narrower molecular weight distribution (*Ð* = 1.63) than in the reaction with the use of Arabica blend’s solution at the same time of the polymerization process. It proves that the Robusta grains have much more antioxidant potential [1]. Both types of solutions received from different blends of coffee grains were analyzed by ^1^H-NMR (Appendix A) to confirm the presence of characteristic compounds, such as caffeine, chlorogenic acids, cafestol, kahweol, and trigonelline [45].

### 2.2. ARGET ATRP of DMAEMA in Various Coffee Extracts Concentration Solutions

Encouraged by the promising effects of homopolymerization of DMAEMA in the solution received from Arabica and Robusta blend, the series of ARGET ATRP reactions were performed at different concentrations of coffee extracts: 5%, 7.5% and 10%, which reflect the percentage of coffee grounds in the solution. The results are summarized in Table 2.

The plots of the natural logarithm of the monomer concentration versus time for each extract used are illustrated in Figure 3a and present a linear dependence, confirming kinetic first-order reactions. Notwithstanding, the use of a higher concentration of coffee extract causes the polymer chains to grow rapidly in a short time, which in turn results in a high viscosity of the reaction mixture, hindering migration of propagating radicals and decreasing the reaction rate. Lowering the concentration of coffee extract to 7.5% increased the reaction rate and higher monomer conversion (Figure 3b). The further reduction in coffee concentration causes a decrease in the reaction rate (compare *k*_p_^app^, Table 2, entries 1–3) due to the slower regeneration of the activator due to low amounts of antioxidants, and consequently resulting in a loss of control over the reaction. However, the molecular weight evolution plot, illustrated in Figure 3b, presents a clean shift toward higher MWs along with the monomer conversion.

### 2.3. ARGET ATRP of Various Monomers

The developed system was also used for the synthesis of other water-soluble polymers, such as poly(oligo(ethylene glycol) methyl ether methacrylate) (POEGMA) and poly(glycidyl methacrylate) (PGMA) (Table 3). The polymerization of these monomers was conducted at the same concentration of coffee extract and amount of catalytic complex. 

The proposed system turned out to be more suitable for the synthesis of short side-chain monomers like GMA and DMAEMA (Figure 4). The polymerizations of the above-mentioned monomers are characterized by a high polymerization rate and a high monomer conversion is reached. Additionally, a polymerization of the hydrophobic *n*-butyl acrylate (*n*BA) in the miniemulsion medium formed from a 10% coffee extract was performed. The reaction followed first order kinetics, but the polymerization rate was low and the received polymer product was characterized by broad molecular weight distribution. This is due to the high concentration of antioxidants in the initial stage of the reaction, which create many initiation sites in reaction with deactivators, which then recombine during the reaction.

### 2.4. Chain-Extension Reaction of PDMAEMA-Br

To investigate the chain-end fidelity, a chain extension experiment of PDMAEMA was performed. The following GPC curves (Figure 5) show chain extension processes of the PDMAEMA-Br macroinitiator with the PDMAEMA second block using Cu^II^Br_2_/TPMA as the catalyst in a 7.5% coffee extract solution. The chain extension reaction was conducted in situ with the following ratio of the reagents: [DMAEMA]_0_/[EBiB]_0_/[Cu^II^Br_2_]_0_/[TPMA]_0_: 150/1/0.01/0.02 in the polymerization of the first block, followed by adding the same volume of the monomer to the reaction mixture in order to incorporate the second block. The PDMAEMA constituting the first block of the copolymers was characterized by a monomodal GPC trace which points to the high retention of chain-end functionality (Figure 5, Appendix A). The second block was successfully incorporated in situ in a controlled manner, proved by linear first-order kinetics (Appendix A) and a shift in the MW peak toward to higher molecular mass (Figure 5).

### 2.5. Electrochemical Measurements of Caffeine Concentration in the Coffee Solutions

The next step of the research focused on the quantitative determination of the caffeine content in the applied coffee solutions using bare GCE (glassy carbon electrode) as a working electrode. For this purpose, an innovative and highly reproducible assay was employed to measure the anodic peak current of caffeine with the use of a differential pulse voltammetry (DPV) technique [46,47]. The calibration curve determined for the concentration range between 0.2 and 1.4 mM (R^2^ = 0.9996, Figure 6) indicated a linear correlation between the anodic peak current and the caffeine concentration in the analyzed solutions, which enables the use of linear regression to calculate the concentration of the examined extracts.

The standard addition method was applied to determine a caffeine concentration in the samples of coffee brews used as polymerization reactions medium. Three independent measurements were made for all points included on the graphs (insets in Figure 7 and Appendix A). DP voltammograms were received for coffee samples diluted 10 times in the supporting electrolyte (0.1 M H_2_SO_4_) and after spiking 200, 400, and 600 µL of the caffeine standard solution with a concentration of 10 mM. Figure 7 reveals an illustrative example of analysis applied for coffee samples with the use of the DPV technique and standard addition method. To achieve additional confirmation for results of DPV analysis, one sample of 7.5% solution of Arabica and Robusta coffee extract and one sample of 7.5% solution of pure Arabica coffee extract were analyzed with the use of a reference HPLC method.

The data presented in Table 4 confirm that results obtained by the DPV analysis are very likely and are within about 5% of the difference in comparison to analysis with the use of the reference HPLC technique.

The caffeine was successfully used as an electron donor in the ARGET ATRP technique, which also confirms reactions carried out in 7.0 mM pure caffeine solution (Appendix A). The antioxidant properties of a coffee were proved by the low dispersity polymer product received in the coffee beverages as in a caffeine solution. In addition, the polymerization rate is influenced by the strength of the coffee brew—the blend of Arabica and Robusta beans showed stronger antioxidant properties compared to the same amount of pure Arabica, which increased the polymerization rate. The weakest antioxidant effect was observed in a pure caffeine solution that does not contain other antioxidant compounds. In this case, the polymerization rate and monomer conversion are the lowest. The different strength of extracts of the same coffee blend undoubtedly contributes to the differences in the rate of polymerization. However, with the increase in the amount of coffee grounds, a continuous increase in caffeine content in coffee beverages is not observed, but only an increase in the number of flavonoids or tannins affects the taste of coffee.

## 3. Materials and Methods

### 3.1. Materials

Coffee of Arabica grains and a mixture of Arabica and Robusta grains were purchased from the Segafredo Zanetti corporation. Ethyl 2-bromoisobutyrate (EBiB, 98%), tetrahydrofuran (THF, >99.9%), acetone (>99.9%), deuterated chloroform (CDCl_3_, >99.8%), hexadecane (HD, 99%), sodium dodecyl sulfate (SDS, 99%), and sodium bromide (NaBr, >99%) were purchased from Sigma Aldrich (St. Louis, MO, USA). Dichloromethane (DCM, >99.5%), sulfuric acid (>95%), and toluene (>99.5%) were purchased from Chempur. *N*,*N*-Dimethylformamide (DMF, 99.9%) and lithium chloride (LiCl, >99%) were purchased from Acros. Anhydrous caffeine (CAF, >99.0%) was purchased from Fluka Analytical. Deuterium oxide (D_2_O, 99.9%) was purchased from Deutero. These reagents were received without further purification. Tris(2-pyridylmethyl)amine (TPMA) was synthesized according to the procedure described in [48]. Cu^II^Br_2_/TPMA stock solutions were prepared according to the reference [49]. Monomers: 2-(dimethylamino)ethyl methacrylate (DMAEMA, >99%; Aldrich), glycidyl methacrylate (GMA, >97.0%; Aldrich), and oligo(ethylene glycol) methyl ether methacrylate (OEGMA_500_; Aldrich) were passed through a column filled with basic alumina prior to use to remove monomethyl ether hydroquinone, which is used as an inhibitor [50].

### 3.2. Analysis

#### 3.2.1. ^1^H-NMR

^1^H-NMR spectra in D_2_O and CDCl_3_ were measured in a Bruker Avance 500 MHz spectrometer (25 °C). Monomer conversion and theoretical number-average molecular weight (*M*_n,th_) were determined by NMR in accordance with the procedures described in [51,52].

#### 3.2.2. GPC

GPC analysis of polymers obtained in solution was performed using a Shimadzu modular system consisting of a CBM-40 system controller, RID-20A differential refractive-index detector, a SIL-20AHT automatic injector, and three PSS GRAM combination columns made of stainless steel (V4A) with pore size: 100 Å, 3000 Å (2 columns), and one precolumn, with the temperature at 30 °C by a CTO-20A oven. The eluent in chromatographic separation was *N*,*N*-Dimethylformamide (HPLC grade with 0.01% *w*/*v* LiCl) with the flow rate set at 1 mL min^−1^. A molecular weight calibration curve was created with the application of monodispersed polystyrene standards (PSS, Polymer Standards Service). The apparent molecular weights (MWs) and molecular weights distributions (MWD) were determined using GPC Postrun Analysis software provided by Shimadzu Corporation.

#### 3.2.3. DPV

Differential pulse voltammetry (DPV) analysis was performed on the Metrohm Autolab potentiostat with the use of a glassy carbon electrode (GCE) (A = 0.071 cm^2^) as the working electrode (WE), a silver chloride electrode with the potential of 0.00 V versus the potential of a saturated calomel electrode (SCE) as the reference electrode (RE) and a platinum wire (l = 7 cm, d = 1 mm) as the counter electrode (CE). The DPV parameters were established as the following: pulse potential of 50 mV, pulse time of 50 ms, and scan rate of 50 mV s^−1^. During the recording of the voltammograms anodic scans were conducted and stored in order to determine the anodic peak current of caffeine, oxidized in the solution of supporting electrolyte (0.1 M H_2_SO_4_ in deionized H_2_O). A calibration curve for caffeine samples was built based on seven different concentrations of caffeine (from 0.2 to 1.4 mM) dissolved in 0.1 M sulfuric acid aqueous solution. For each point of the curve, three independent DPV scans were done and as a value of the point the mean of the current values for these three measurements was approved. The DPV analysis of coffee brews was performed using the standard addition method for samples of coffee extracts diluted at the ratio of 1:10 in 0.1 M H_2_SO_4_ aqueous solution followed by spiking 200, 400, and 600 µL of 10 mM caffeine standard solution in 0.1 M H_2_SO_4_. For each addition, three independent DPV scans were recorded and as a result the mean of the current values was adopted.

#### 3.2.4. HPLC

High performance liquid chromatography (HPLC) analysis was performed using a 1290 Infinity LC system with a DAD detector, equipped with an Agilent ZORBAX Eclipse Plus C18, Rapid Resolution HT, 90A, (4.6 × 50 mm, 1.8 µm, 600 bar) column (thermostated at 30 °C). The analysis of the chromatograms was done with the use of LC OpenLab (Agilent Technologies, Waldbronn, Germany). The experimental conditions were the following: the mobile phase was water (75%) and methanol (25%) solution, flow rate of 1.0 mL min^−1^, and caffeine was detected by monitoring the absorbance at wavelength of 272 nm. The calibration curve was established for caffeine samples in a range of concentration from 0.05 to 1.68 mM, prepared by dissolving caffeine in the mobile phase. A good linearity was found within the entire concentration range studied. For coffee sample analysis, beverages were diluted at the ratio of 1:20 in the mobile phase. Thus, received solutions were filtered on amine tubes (1 mL) and through 0.45 μm nylon filters (Agilent Technologies) prior to analysis. The concentration of CAF was determined using the calibration curve previously described. For each sample HPLC analysis was performed two times, and as a result, the mean of these two values was adopted. The calibration curve equation is reported in point S.8 in Appendix A.

## 4. Conclusions

The polymerization of hydrophilic and hydrophobic monomers in coffee extracts was examined by the ARGET ATRP technique. For this purpose, solutions of Arabica coffee beans extracts and an equally proportionate mixture of Arabica and Robusta beans extracts were tested. A blend with Robusta beans, which contains a high number of antioxidants, turns out to be an excellent reaction medium. Coffee extract solution with a caffeine concentration of 7.0 mM obtained homopolymers in good yields and in a quite short time. Additionally, the chain-end functionality was confirmed by an in situ chain extension reaction, which showed a clean shift in the MW peak toward a higher molecular mass. The coffee solution is also an appropriate reaction medium for the preparation of hydrophobic homopolymers in a miniemulsion system using the ARGET ATRP technique. Polymerizations carried out in coffee solutions not only confirm the presence of antioxidants in the coffee solution, but also provide real possibilities of using the waste generated during the processing of coffee beans. In addition, the mild polymerization environment guarantees the responsible use of PDMAEMA brushes as cationic antibacterial surfaces and nonviral gene carriers in biomedical applications without diligent polymer purification.

## Figures and Tables

**Figure 1 molecules-27-00840-f001:**
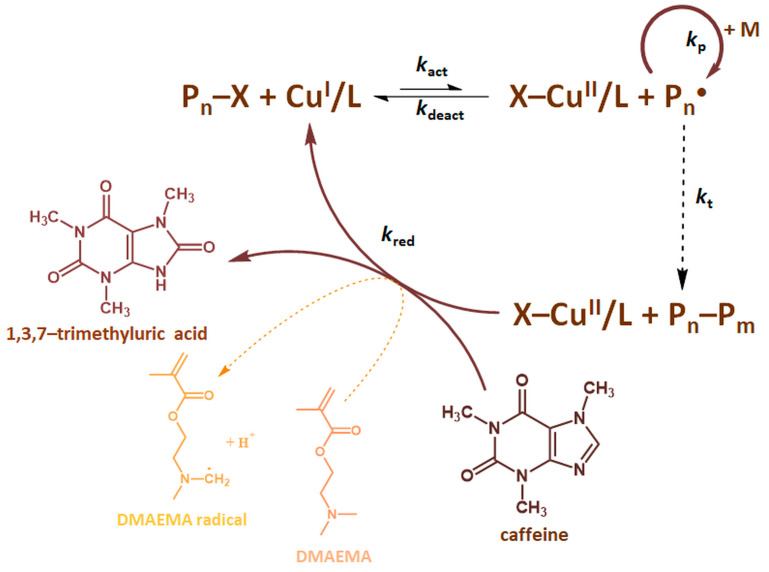
The mechanism of ARGET ATRP in a coffee solution.

**Figure 2 molecules-27-00840-f002:**
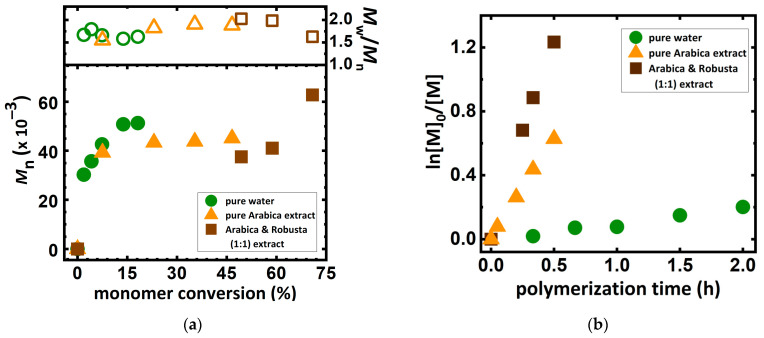
(**a**) Semilogarithmic kinetic plot, (**b**) *M*_n_ and *M*_w_/*M*_n_ *vs.* monomer conversion for ARGET ATRP in preparation of PDMAEMA in various solvents (Table 1). Reactions conditions as in Table 1.

**Figure 3 molecules-27-00840-f003:**
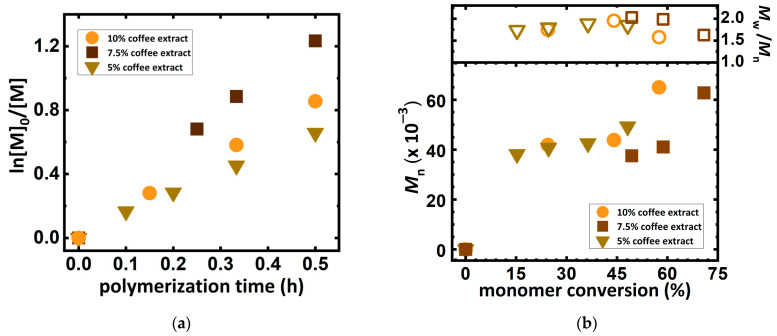
(**a**) First-order kinetic plot of DMAEMA conversion versus polymerization time for different coffee extracts according to Table 2, entries 1–3. (**b**) *M*_n_ and *M*_w_/*M*_n_ *vs.* monomer conversion according to Table 2, entries 1–3.

**Figure 4 molecules-27-00840-f004:**
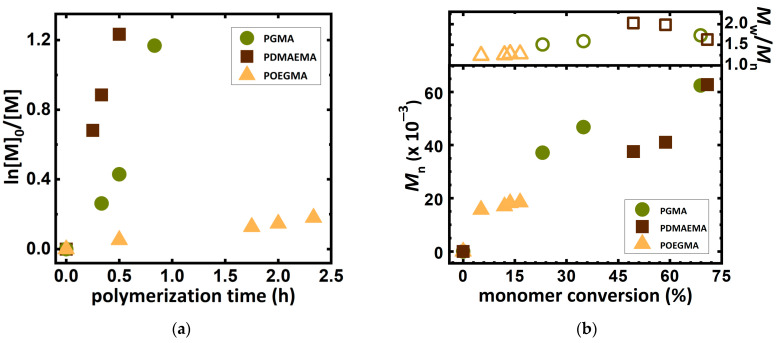
(**a**) First-order kinetic plot of monomer conversion versus polymerization time according to Table 3, entries 1–3. (**b**) *M*_n_ and *M*_w_/*M*_n_ *vs.* monomer conversion according to Table 3, entries 1–3.

**Figure 5 molecules-27-00840-f005:**
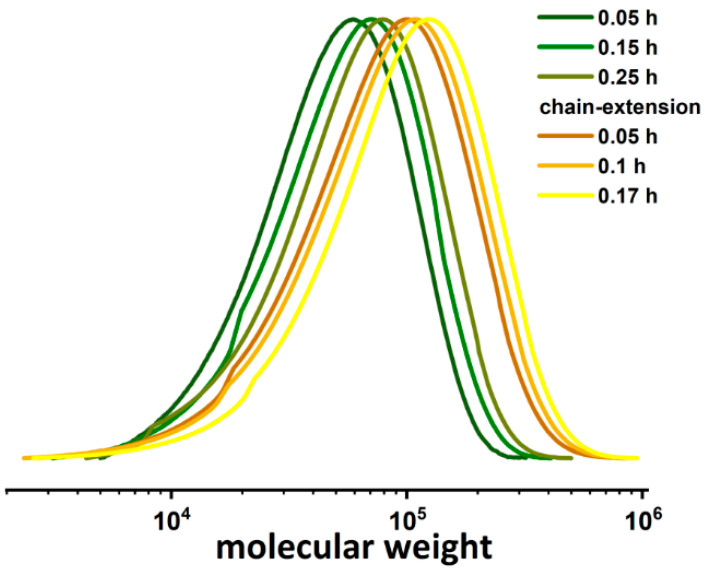
GPC results of PDMAEMA-Br chain-extension reaction.

**Figure 6 molecules-27-00840-f006:**
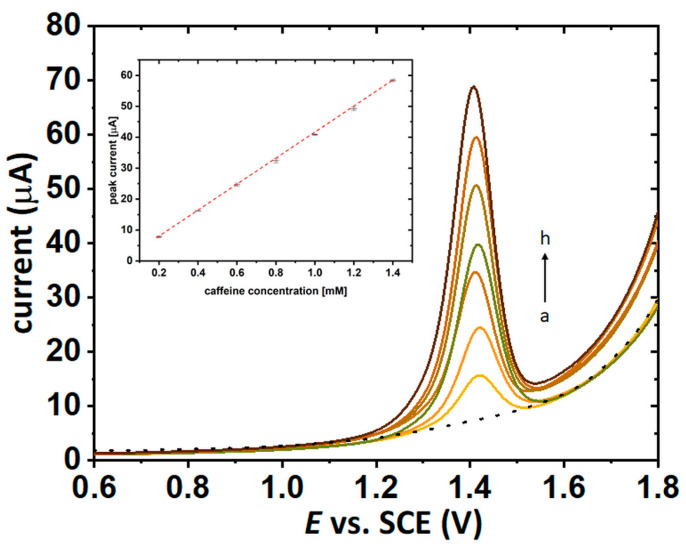
DP voltammograms received for various caffeine concentration: (a) 0.0, (b) 0.2, (c) 0.4, (d) 0.6, (e) 0.8, (f) 1.0, (g) 1.2, and (h) 1.4 mM in 0.1 M H_2_SO_4_ aqueous solution on GCE. The linear dependence between the peak current and caffeine concentration is appended in the inset. DPV parameters: pulse potential of 50 mV, pulse time of 50 ms and scan rate of 50 mV/s.

**Figure 7 molecules-27-00840-f007:**
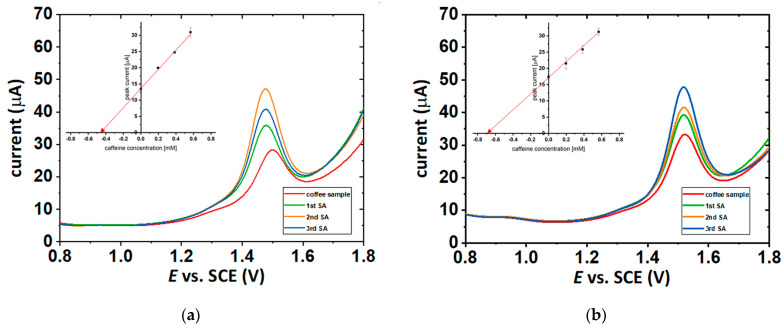
DP voltammograms received on GCE in the analysis of coffee samples in 0.1 M H_2_SO_4_ aqueous solution using the standard addition method with 1 mL of the sample and after spiking 200, 400, and 600 µL of 10 mM caffeine standard solution, resulting in (**a**) 7.5% Arabica sample and (**b**) 7.5% Arabica and Robusta (50/50%) sample. The analysis, by standard addition method, is presented in the insets. DPV parameters: pulse potential of 50 mV, pulse time of 50 ms, and scan rate of 50 mV/s.

**Table 1 molecules-27-00840-t001:** ARGET ATRP of DMAEMA in various coffee bean blend solutions.

Entry ^1^	Type of Coffee ^2^	Conv ^3^ (%)	DP_app_ ^3^	*k*_p_^app 3^ (h^−1^)	*M*_n,th_^4^ (×10^−3^)	*M*_n,app_^5^ (×10^−3^)	*Ð* ^5^	*f*^6^ (%)
1	pure water	18.2	36	0.094	5.9	51.3	1.63	863
2	Arabica & Robusta (50/50%)	70.9	142	2.55	22.6	62.8	1.63	278
3	Arabica (100%)	46.6	93	1.28	14.9	45.1	1.88	302

^1^ General reaction conditions: [DMAEMA]_0_/[EBiB]_0_/[Cu^II^Br_2_]_0_/[TPMA]_0_: 200/1/0.01/0.02; T = 22 °C; V_tot_ = 8 mL (DMAEMA/solvent = 0.4/0.6 by *v*/*v*), [DMAEMA]_0_ = 19 mM, [I]_0_ = 0.09 mM. [NaBr] = 0.1 M, [Cu^II^Br_2_]_0_ = 1.19 µM (2-fold excess of TPMA), t = 0.5 h, except entry 1: t = 2 h. ^2^ Mixture of Arabica and Robusta (50/50%) and Arabica 100% were purchased from Segafredo Zanetti corporation as Espresso CASA and Arabica coffee, respectively. ^3^ Monomer conversion, polymerization rate (*k*_p_^app^), and DP_app_ were determined by ^1^H-NMR. ^4^ *M*_n,th_ = ([M]_0_/[I]_0_) × Conv × *M*_monomer_ + *M*_initiator_. ^5^ *M*_n,app_ and *Ð* was determined by DMF + 10 mM LiCl GPC with PS standards. ^6^ Initiation efficiency calculated as: *f* = *M*_n,app_/*M*_n,th_.

**Table 2 molecules-27-00840-t002:** ARGET ATRP of DMAEMA in various coffee extract concentration solutions.

Entry ^1^	Concentration of Coffee Extract ^2^ (%)	Conv ^3^ (%)	DP_app_ ^3^	*k*_p_^app 3^ (h^−1^)	*M*_n,th_^4^ (×10^−3^)	*M*_n,app_^5^ (×10^−3^)	*Ð* ^5^
1	10	57.5	115	1.73	18.4	65.0	1.58
2	7.5	70.9	142	2.55	22.6	62.8	1.63
3	5	48.2	96	1.34	15.4	49.3	1.85

^1^ General reaction conditions: [DMAEMA]_0_/[EBiB]_0_/[Cu^II^Br_2_]_0_/[TPMA]_0_: 200/1/0.01/0.02; T = 22 °C; V_tot_ = 8 mL (DMAEMA/solvent = 0.4/0.6 by *v*/*v*), [DMAEMA]_0_ = 19 mM, [I]_0_ = 0.09 mM. [NaBr] = 0.1 M, [Cu^II^Br_2_]_0_ = 1.19 µM (2-fold excess of TPMA), t = 0.5 h. ^2^ Mixture of Arabica and Robusta (50/50%) and Arabica 100% were purchased from Segafredo Zanetti corporation as Espresso CASA and Arabica coffee, respectively. ^3^ Monomer conversion, polymerization rate (*k*_p_^app^), and DP_app_ were determined by ^1^H-NMR. ^4^ *M*_n,th_ = ([M]_0_/[I]_0_) × Conv × *M*_monomer_ + *M*_initiator_. ^5^ *M*_n,app_ and *Ð* was determined by DMF + 10 mM LiCl GPC with PS standards.

**Table 3 molecules-27-00840-t003:** ARGET ATRP of various monomers in 7.5% solution received from an equally proportional blend of Arabica and Robusta beans.

Entry ^1^	Monomer	t (h)	Conv ^2^ (%)	DP_app_ ^3^	*k*_p_^app 2^ (h^−1^)	*M*_n,th_^3^ (×10^−3^)	*M*_n,app_^4^ (×10^−3^)	*Ð* ^4^
1	DMAEMA	0.5	70.9	142	2.55	22.6	62.8	1.63
2	GMA	1	79.1	158	1.38	22.7	69.5	1.78
3	OEGMA	2.33	16.5	33	0.08	16.7	18.5	1.28

^1^ General reaction conditions: [Monomer]_0_/[EBiB]_0_: 200/1; [Cu^II^Br_2_] = 9.5 ppm by wt (2-fold excess of TPMA); [NaBr] = 0.1 M, T = 22 °C; V_tot_ = 8 mL (monomer/solvent = 0.4/0.6 by *v*/*v*). ^2^ Monomer conversion, polymerization rate (*k*_p_^app^), and DP_app_ were determined by ^1^H-NMR. ^3^ *M*_n,th_ = ([M]_0_/[I]_0_) × Conv × *M*_monomer_ + *M*_initiator_. ^4^ *M*_n,app_ and *Ð* was determined by DMF + 10 mM LiCl GPC with PS standards.

**Table 4 molecules-27-00840-t004:** Comparison of caffeine concentration analysis results received for various samples of coffee brews with the use of DPV and HPLC techniques.

Coffee Solution	Caffeine Concentration (mM)	
	DPV Analysis	Reference HPLC
7.5% Arabica	4.47 (s *: 7.64 × 10^−7^)	4.31 (s: 7.28 × 10^−2^)
5% Arabica and Robusta (50/50%)	4.65 (s: 1.20 × 10^−6^)	-
7.5% Arabica and Robusta (50/50%)	7.00 (s: 1.14 × 10^−6^)	7.08 (s: 1.56 × 10^−2^)
10% Arabica and Robusta (50/50%)	7.23 (s: 1.35 × 10^−6^)	-

* s: standard deviation.

## Data Availability

All data and supporting data are reported in this manuscript and Appendix A.

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
