# Peer review of "Coffee Beverage: A New Strategy for the Synthesis of Polymethacrylates via ATRP"

_molecules, 2022, doi:10.3390/molecules27030840_

Round 1
Reviewer 1 Report
Polymerization of hydrophilic monomers at room temperature via ARGET ATRP wherein high antioxidant (caffeine) containing coffee solution (pure Arabica; Robusta & Arabica (1:1)) is used. However, being a mixture of organic compounds, there is no mention of the mechanism of the polymerization.
What is the actual mechanism of the polymerization? How is the initiation taking place? Which species is involved in controlling the polymerization?
Author Response
Manuscript number: molecules-1529687
Coffee beverage: A new strategy for the synthesis of polymethacrylates via ATRP
We thank the reviewers for their valuable comments and suggestions. All changes and amendments are highlighted in the new text. Our point-by-point response follows:
Response to Reviewer 1:
Comment 1: Polymerization of hydrophilic monomers at room temperature via ARGET ATRP wherein high antioxidant (caffeine) containing coffee solution (pure Arabica; Robusta & Arabica (1:1)) is used. However, being a mixture of organic compounds, there is no mention of the mechanism of the polymerization.
As suggested, the detailed discussion about mechanism of polymerization was improved. The detailed mechanism of ARGET ATRP, in which the reducing agent was caffeine constituing the main antioxidant component of coffee, was presented at page 3. Voltammetric measurements were carried out, which confirmed the strongest antioxidant effect of caffeine in both mixtures, therefore the mechanism in which caffeine is the dominant antioxidant factor controlling the polymerization process was included in the manuscript at page 3. Nevertheless, on DP voltammograms occure a slight signal derived from the additional antioxidants, i.e. chlorogenic acids, cafestol, kahweol and trigonelline contained in the coffee extract, which also help to reduce the deactivator to its active form, but they are not the main reducing substances. It should also be emphasized that an additional confirmation of the effect of caffeine as the main reducing agent is the mass of polymer obtained in a pure caffeine solution (MW: 49100) comparable to the mass of polymer obtained in a solution of a coffee blend: in pure Arabica extract MW: 45100; in Arabica & Robusa (1:1) extract MW: 62800
An additional information was added in the Results of the Main text (page 3, highlighted version).
Comment 2: What is the actual mechanism of the polymerization?
The actual mechanism of the polymerization is according to ARGET ATRP with caffeine as main reducing agent, confirmed with the voltammetry measurements. Caffeine, being the main antioxidant present in coffee, is oxidized in the reaction environment with the simultaneous incorporation of oxygen derived from the reaction solvent. The process takes place in a mild oxidizing medium of copper(II) bromide, where, as in the permanganate environment, a two-electron redox process takes place for the caffeine molecule. The mechanism involves the interaction of the oxidant - deactivator with the caffeine molecule to give a free radical derived from caffeine. Then, water particles from the reaction solvent react with a free radical, giving 8-hydroxy caffeine which fast tautomerise to the final oxidation product of caffeine - 1,3,7-trimethyluric acid (Figure 1 at main manuscript). According to the ATRP mechanism, the dynamic equilibrium of the copper(I) and copper(II) complex is involved in controlling polymerization. The complex of copper(II) introduced into the polymerization system undergoes reduction due to the caffeine action. Additionally, trace amounts of oxygen could cause oxidation of the copper and affect the dynamic equilibrium of CuII to CuI estabilishing. The polymerization process is then controlled by appropriate ratio of activator to deactivator species.
An additional comment was added in the Results of the Main text (page 3, highlighted version).
Comment 3: How is the initiation taking place?
The initiation taking place as presented at page 2 of manuscript: „mechanism of a coffee–induced ATRP based on radicals generation process involving a copper−amine complex (activator) in its lower oxidation state, CuI/L, is illustrated in Figure 1. The catalytic complex activates an alkyl halide initiator as ethyl 2-bromoisobutyrate (EBiB) or dormant chain end (Pn−X), forming propagating radicals and oxidizing to a higher oxidation state.”
Information can be found in the Results of the Main text (page 3, highlighted version).
Comment 4: Which species is involved in controlling the polymerization?
According to the ATRP mechanism, the dynamic equilibrium of the copper(I) and copper(II) complex is involved in controlling polymerization. The complex of copper(II) introduced into the polymerization system undergoes reduction due to the caffeine action. Additionally, trace amounts of oxygen could cause oxidation of the copper and affect the dynamic equilibrium of CuII to CuI estabilishing. The polymerization process is then controlled by appropriate ratio of activator to deactivator species.
An additional comment was added in the Results of the Main text (page 3, highlighted version).
Additionally, English language and style was checked carefully. All corrections have been made as follows:
Before correction |
Corrected version in revised text |
Page of the revised text (highlighted version) |
Coffee − the most popular beverage in the 21st-century society … |
Coffee − the most popular beverage in the 21st- 21st century society |
1 |
|
|
|
pure Arabica beans and proportional blend of Arabica and Robusta beans |
pure Arabica beans and a proportional blend of Arabica and Robusta beans |
1 |
…significant antioxidant capacity |
…significant antioxidant activitycapacity |
1 |
The highest amount of compounds with antioxidant potential was proved in medium roast coffee due to..… |
The highest amount of compounds with antioxidant potential was determinedproved in medium roast coffee grains due to… |
1 |
There are over five hundred types of coffee |
There are over five hundred typesspecies of coffee |
1 |
So far, mainly vitamins such as ascorbic acid [20-23] or riboflavin [12, 24] and sugars such as glucose [25] and cyclodextrin [26]. |
So far, mainly used vitamins such as ascorbic acid [20-23] or riboflavin [12, 24] and sugars such as glucose [25] and cyclodextrin [26], but also the novel external reducing agents as liquid metal [27] were used. |
2 |
Equally important is the solubility of the monomers in the individual media, which affects the choice of the reaction solvent. |
Equally important is the solubility of the monomers in the individual media, which determines the selectionaffects the choice of the reaction solvent. |
2 |
Antioxidants derived from a coffee extract as caffeine, chlorogenic acids, cafestol, kahweol and trigonelline trigger a reduction of a deactivator complex, X-CuII/L, ensuring continuous catalytic reaction. |
Antioxidants derived from a coffee extract such as caffeine, chlorogenic acids, cafestol, kahweol and trigonelline trigger a reduction of a deactivator complex, X-CuII/L, ensuring continuous catalytic reaction. Voltammetric measurements were carried out, which confirmed the strongest antioxidant effect of caffeine in both mixtures, therefore the mechanism in which caffeine is the dominant antioxidant factor controlling the polymerization process was presented. |
3 |
Caffeine, being the main antioxidant of coffee,… |
Caffeine, being the main antioxidant present inof coffee,… |
3 |
The process takes place in a mild oxidizing environment of copper (II) bromide, … |
The process takes place in a mild oxidizing environmentmedium of copper(II) bromide, … |
3 |
The mechanism involves the attack of the oxidant - deactivator on the caffeine molecule to give a free radical derived from caffeine. |
The mechanism involves the attackinteraction of the oxidant - deactivator onwith the caffeine molecule to give a free radical derived from caffeine. |
3 |
Then, water particles from the reaction solvent react with a free radical, giving 8-hydroxy caffeine which fast tautemerise to the final oxidation product of caffeine - 1,3,7-trimethyluric acid. [40] |
Then, water particles from the reaction solvent react with a free radical, giving 8-hydroxy caffeine which fast tautemerisetautomerise to the final oxidation product of caffeine - 1,3,7-trimethyluric acid (Figure 1) [40]. According to the ATRP mechanism, the dynamic equilibrium of the copper(I) and copper(II) complex is involved in controlling polymerization. The complex of copper(II) introduced into the polymerization system undergoes reduction due to the caf-feine action. Additionally, trace amounts of oxygen could cause oxidation of the copper and affect the dynamic equilibrium of CuII to CuI estabilishing. The polymerization process is then controlled by appropriate ratio of activator to deactivator species [8, 23]. |
3 |
Therefore, all next polymerization processes were performed at room temperature, allowing to achieve products in controlled manner and in a shorter time. For the synthesis in pure water without the addition of a reducing agent |
Therefore, all next polymerization processes were performed at room temperature, al-lowing to achieve products in controlled manner and in a shorter time. In efficient ATRP catalyst system, metal center must have two easily accessible oxidation states as well as must be able to accept the halogen, which requires a suffi-ciently high halogenophilicity [41]. Low halidophilicity i.e. the association of the halide an-ion to the catalyst in the higher oxidation state, related to the connection of the halo-gen ion create possibility of disproportionation of CuI/L to X-CuII/L and Cu0 [42, 43]. Finally, halide anions must have high affinity for CuIIL2+, such as to provide the pres-ence of enough concentration of deactivator to give a well-controlled process. In aqueous systems, the X-CuII/L bond can easily dissociate to CuIIL2+ and Br−, typically required is high concentration of catalyst is needed or the presence of salts with halide anions in order to shift the equilibrium toward the X-CuII/L deactivator species [28, 44]. Because of experiments were carried out in low ppm concentration of catalyst (9.5 ppm by wt%), the effect of the addition of a halide salt on the polymerization process in the medium of a coffee blend extract was investigated (Arabica & Robusta 1:1). Indeed, the lack of a halide salt caused a loss of process control due to ineffective catalyst regeneration, the addition of NaBr resulted in an increase of control over polymerization reflected valid initiation efficiency values and a narrower molecular weight distribution (Table S2). Higher salt concentration affected more effective catalyst re-generation and an increase in control over the process (compare initiation efficiency in Table S2), however, the reaction stopped after 0.5 hour did not reach a satisfactory MWD (Figure S3b), therefore it was decided to conduct subsequent reactions at a sodium bromide concentration equal to 0.1 M. For the synthesis in pure water without the addition of a reducing agent |
4 |
On the other hand, the N-CH2 group remains the initiation site of radical polymerization, resulting in a loss of ATP control as indicated by an almost 900% initiation efficiency. |
On the other hand, the N-CH2 group remains the initiation site of radical polymerization, resulting in a loss of ATPATRP control as indicated by an almost 900% initiation efficiency. |
4 |
Table 1. footer: 1 General reaction conditions: [DMAEMA]0/[EBiB]0/[CuIIBr2]0/[TPMA]0: 200/1/0.01/0.02; T = 22°C; Vtot = 8 mL (DMAEMA/solvent = 0.4/0.6 by v/v), [DMAEMA]0 = 19 mM, [I]0 = 0.09 mM. [CuIIBr2]0 = 1.19 µM (2-fold excess of TPMA), t = 0.5 h, except entry 1: t = 2h.
|
Table 1. footer: 1 General reaction conditions: [DMAEMA]0/[EBiB]0/[CuIIBr2]0/[TPMA]0: 200/1/0.01/0.02; T = 22°C; Vtot = 8 mL (DMAEMA/solvent = 0.4/0.6 by v/v), [DMAEMA]0 = 19 mM, [I]0 = 0.09 mM. [NaBr] = 0.1 M, [CuIIBr2]0 = 1.19 µM (2-fold excess of TPMA), t = 0.5 h, except entry 1: t = 2h.
|
5 |
caused by low amount of catalytic complex – 9.5 ppm by wt%, obtained PDMAEMA |
caused by low amount of catalytic complex – 9.5 ppm by wt%, obtained PDMAEMA |
5 |
Both types of solutions received from different blends of coffee grains were analyzed by 1H NMR (Figure S4) to confirm |
Both types of solutions received from different blends of coffee grains were analyzed by 1H NMR (Figure S4Figure S5) to confirm |
5 |
Table 2. footer: 1 General reaction conditions: [DMAEMA]0/[EBiB]0/[CuIIBr2]0/[TPMA]0: 200/1/0.01/0.02; T = 22°C; Vtot = 8 mL (DMAEMA/solvent = 0.4/0.6 by v/v), [DMAEMA]0 = 19 mM, [I]0 = 0.09 mM. [CuIIBr2]0 = 1.19 µM (2-fold excess of TPMA), t = 0.5 h.
|
Table 2. footer: 1 General reaction conditions: [DMAEMA]0/[EBiB]0/[CuIIBr2]0/[TPMA]0: 200/1/0.01/0.02; T = 22°C; Vtot = 8 mL (DMAEMA/solvent = 0.4/0.6 by v/v), [DMAEMA]0 = 19 mM, [I]0 = 0.09 mM. [NaBr] = 0.1 M, [CuIIBr2]0 = 1.19 µM (2-fold excess of TPMA), t = 0.5 h.
|
6 |
Table 3. footer: 1General reaction conditions: [Monomer]0/[EBiB]0: 200/1; [CuIIBr2]= 9.5 ppm by wt (2-fold excess of TPMA); T = 22°C; Vtot = 8 mL (monomer/solvent = 0.4/0.6 by v/v).
|
Table 3. footer: 1General reaction conditions: [Monomer]0/[EBiB]0: 200/1; [CuIIBr2]= 9.5 ppm by wt (2-fold excess of TPMA); [NaBr] = 0.1 M, T = 22°C; Vtot = 8 mL (monomer/solvent = 0.4/0.6 by v/v).
|
7 |
In order to investigate the chain-end fidelity, chain extension experiment of PDMAEMA were performed. |
In order to investigate the chain-end fidelity, chain extension experiment of PDMAEMA werewas performed. |
8 |
The PDMAEMA constituting the first block of the copolymers was characterized by monomodal GPC trace which points to the high retention of chain-end functionality (Figure 5, Figure S12). |
The PDMAEMA constituting the first block of the copolymers was characterized by monomodal GPC trace which points to the high retention of chain-end functionality (Figure 5, Figure S1213). |
8 |
The second block was successfully incorporated in situ in a controlled manner proved by linear first-order kinetics (Figure S13) and a shift in the MW peak toward to higher molecular mass (Figure 5). |
The second block was successfully incorporated in situ in a controlled manner proved by linear first-order kinetics (Figure S1314) and a shift in the MW peak toward to higher molecular mass (Figure 5). |
8 |
Three independent measurements were made for the all points included on the graphs (insets in the Figure 7 and Figure S15). |
Three independent measurements were made for the all points included on the graphs (insets in the Figure 7 and Figure S1516). |
9 |
The caffeine was successfully used as an electron donor in the ARGET ATRP technique, which also confirms reaction carried out in 7.0 mM pure caffeine solution (Table S4). |
The caffeine was successfully used as an electron donor in the ARGET ATRP technique, which also confirms reaction carried out in 7.0 mM pure caffeine solution (Table S4Table S5). |
10 |
The calibration curve equation is reported in point S.8.
|
The calibration curve equation is reported in point S.8 in Supplementary Materials.
|
12 |
Table S1: ARGET ATRP of DMAEMA in various temperature, Figure S1: (a) Semilogarithmic kinetic plot, (b) Mn and Mw/Mn vs. monomer conversion for preparation of PDMAEMA in coffee solution at various temperature (Table S1, entries 1-2). Reaction conditions as in Table S1, Figure S2: GPC traces of PDMAEMA chains prepared according to (a) Table S1 entry 1 and (b) Table S1 entry 2 using as eluent DMF + 10 mM LiCl, Figure S3: GPC traces of PDMAEMA chains prepared according to (a) Table 1 entry 1; (b) Table 1 entry 2 and (c) Table 1 entry 3 using as eluent DMF + 10 mM LiCl, Figure S4: 1H NMR stack plot of coffee samples: Segafredo Arabica and Segafredo Espresso CASA in D2O (500 MHz, 25°C), Figure S5: GPC traces of PDMAEMA chains prepared according to (a) Table 2 entry 1; (b) Table 2 entry 2 and (c) Table 2 entry 3 using as eluent DMF + 10 mM LiCl, Figure S6: GPC traces of (a) PDMAEMA; (b) PGMA; (c) POEGMA chains prepared according to (a) Table 3 entry 1; (b) Table 3 entry 2 and (c) Table 3 entry 3 using as eluent DMF + 10 mM LiCl, Table S2: ARGET ATRP of n-BA in miniemulsion medium of 10% coffee solution, Figure S7: Polymerization of n-BA in miniemulsion system: (a) First-order kinetic plot of monomer conversion vs. polymerization time. (b) Mn and Mw/Mn vs. monomer conversion. (c) GPC traces of n-BA chains with THF as the eluent, Figure S8: 1H NMR spectrum of PDMAEMA homopolymer (after purification) in CDCl3 (500 MHz, 25°C). The chemical shifts characteristic for PDMAEMA chains were assigned: δ (ppm) = 0.72-1.15 (3H, –CH3, α), 1.72-2.07 (2H, –CH2–, β), 2.23-2.53 (6H, 2 x -CH3, c), 2.57-2.84 (2H, –CH2–, b) and 3.99-4.19 (2H, –CH2–, a), Figure S9: 1H NMR spectrum of PGMA homopolymer (after purification) in CDCl3 (500 MHz, 25°C). The chemical shifts characteristic for PGMA chains were assigned: δ (ppm) = 0.80-1.18 (3H, –CH3, α), 1.84-2.06 (2H, –CH2–, β), 2.56-2.96 (2H, –CH2–, c), 3.15-3.34 (1H, –CH–, b), 3.71-3.89 and 4.19-4.39 (2H, –CH2–, a), Figure S10: 1H NMR spectrum of POEGMA homopolymer (after purification) in CDCl3 (500 MHz, 25°C). The chemical shifts characteristic for POEGMA chains were assigned: δ (ppm) = 0.78-0.97 (3H, –CH3, α), 1.23-1.30 (2H, –CH2–, β), 3.34-3.44 (3H, CH3–, c), 3.54-3.58 (2H, –CH2–, b-first segment), 3.59-3.82 (28H, –CH2– and –CH2–, a + b), and 4.01-4.17 (2H, –CH2–, a-first segment), Figure S11: 1H NMR spectrum of PBA homopolymer (after purification) in CDCl3 (500 MHz, 25°C). The chemical shifts characteristic for PBA chains were assigned: δ (ppm) = 0.85-1.01 (3H, –CH3, d), 1.31-2.00 (6H, –CH2–, a + b + c), 2.20-2.43 (1H, –CH=, α) and 3.90-4.14 (2H, –CH2–, β), Table S3: Chain-extension reaction of PDMAEMA in 7.5% coffee solution, Figure S12: Polymerization of PDMAEMA (first block) in 7.5% coffee solution (a) First-order kinetic plot of monomer conversion vs. polymerization time. (b) Mn and Mw/Mn vs. monomer conversion, Figure S13: Copolymerization of PDMAEMA (second block) in 7.5% coffee solution (a) First-order kinetic plot of monomer conversion vs. polymerization time. (b) Mn and Mw/Mn vs. monomer conversion, Figure S14: The calibration curve received for caffeine determination by DPV technique and the relative equation above the curve. The curve depicts linear dependance between the anodic peak current and caffeine concentration in 0.1 M H2SO4 aqueous solution. DP voltammograms received on GCE. DPV parameters: pulse potential of 50 mV, pulse time of 50 ms and scan rate of 50 mV/s, Figure S15: DP voltammograms received on GCE in analysis of coffee samples in 0.1 M H2SO4 aqueous solution using the standard addition method with 1 mL of the sample and after spiking 200, 400 and 600 µL of 10 mM caffeine standard solution, (a) 5% Arabica & Robusta (50/50%) sample; (b) 10% Arabica & Robusta (50/50%) sample. The analysis by standard addition method is presented in the insets. DPV parameters: pulse potential of 50 mV, pulse time of 50 ms and scan rate of 50 mV/s, Figure S16: The calibration curve received for caffeine determination by HPLC technique and the relative equation on the side. Analysis performed with the use of 1290 Infinity LC system with DAD detector and Agilent ZORBAX Eclipse Plus C18, Rapid Resolution HT, 90A, (4.6 x 50 mm, 1.8 um, 600 bar) column in 30°C. Mobile phase: water (75%) and methanol (25%), flow rate: 1.0 ml/min. Wavelength used for caffeine detection: 272 nm, Figure S17. Polymerization of DMAEMA in 7.0 mM caffeine solution: (a) First-order kinetic plot of monomer conversion vs. polymerization time. (b) Mn and Mw/Mn vs. monomer conversion. (c) GPC traces of DMAEMA with DMF + 10 mM LiCl as the eluent. |
Table S1: ARGET ATRP of DMAEMA in various temperature, Figure S1: (a) Semilogarithmic kinetic plot, (b) Mn and Mw/Mn vs. monomer conversion for preparation of PDMAEMA in coffee solution at various temperature (Table S1, entries 1-2). Reaction conditions as in Table S1, Figure S2: GPC traces of PDMAEMA chains prepared according to (a) Table S1 entry 1 and (b) Table S1 entry 2 using as eluent DMF + 10 mM LiCl, Table S2. ARGET ATRP in various NaBr concentration in 7.5% solution received from equally proportional blend of Arabica and Robusta beans, Figure S3. (a) Semilogarithmic kinetic plot, (b) Mn and Mw/Mn vs. monomer conversion for ARGET ATRP in preparation of PDMAEMA in various NaBr concentration (Table S2). Reactions conditions as in Table S2. Figure S3: GPC traces of PDMAEMA chains prepared according to (a) Table 1 entry 1; (b) Table 1 entry 2 and (c) Table 1 entry 3 using as eluent DMF + 10 mM LiCl, Figure S4: 1H NMR stack plot of coffee samples: Segafredo Arabica and Segafredo Espresso CASA in D2O (500 MHz, 25°C), Figure S5: GPC traces of PDMAEMA chains prepared according to (a) Table 2 entry 1; (b) Table 2 entry 2 and (c) Table 2 entry 3 using as eluent DMF + 10 mM LiCl, Figure S6: GPC traces of (a) PDMAEMA; (b) PGMA; (c) POEGMA chains prepared according to (a) Table 3 entry 1; (b) Table 3 entry 2 and (c) Table 3 entry 3 using as eluent DMF + 10 mM LiCl, Table S2: ARGET ATRP of n-BA in miniemulsion medium of 10% coffee solution, Figure S7: Polymerization of n-BA in miniemulsion system: (a) First-order kinetic plot of monomer conversion vs. polymerization time. (b) Mn and Mw/Mn vs. monomer conversion. (c) GPC traces of n-BA chains with THF as the eluent, Figure S8: 1H NMR spectrum of PDMAEMA homopolymer (after purification) in CDCl3 (500 MHz, 25°C). The chemical shifts characteristic for PDMAEMA chains were assigned: δ (ppm) = 0.72-1.15 (3H, –CH3, α), 1.72-2.07 (2H, –CH2–, β), 2.23-2.53 (6H, 2 x -CH3, c), 2.57-2.84 (2H, –CH2–, b) and 3.99-4.19 (2H, –CH2–, a), Figure S9: 1H NMR spectrum of PGMA homopolymer (after purification) in CDCl3 (500 MHz, 25°C). The chemical shifts characteristic for PGMA chains were assigned: δ (ppm) = 0.80-1.18 (3H, –CH3, α), 1.84-2.06 (2H, –CH2–, β), 2.56-2.96 (2H, –CH2–, c), 3.15-3.34 (1H, –CH–, b), 3.71-3.89 and 4.19-4.39 (2H, –CH2–, a), Figure S10: 1H NMR spectrum of POEGMA homopolymer (after purification) in CDCl3 (500 MHz, 25°C). The chemical shifts characteristic for POEGMA chains were assigned: δ (ppm) = 0.78-0.97 (3H, –CH3, α), 1.23-1.30 (2H, –CH2–, β), 3.34-3.44 (3H, CH3–, c), 3.54-3.58 (2H, –CH2–, b-first segment), 3.59-3.82 (28H, –CH2– and –CH2–, a + b), and 4.01-4.17 (2H, –CH2–, a-first segment), Figure S11: 1H NMR spectrum of PBA homopolymer (after purification) in CDCl3 (500 MHz, 25°C). The chemical shifts characteristic for PBA chains were assigned: δ (ppm) = 0.85-1.01 (3H, –CH3, d), 1.31-2.00 (6H, –CH2–, a + b + c), 2.20-2.43 (1H, –CH=, α) and 3.90-4.14 (2H, –CH2–, β), Table S3: Chain-extension reaction of PDMAEMA in 7.5% coffee solution, Figure S12: Polymerization of PDMAEMA (first block) in 7.5% coffee solution (a) First-order kinetic plot of monomer conversion vs. polymerization time. (b) Mn and Mw/Mn vs. monomer conversion, Figure S13: Copolymerization of PDMAEMA (second block) in 7.5% coffee solution (a) First-order kinetic plot of monomer conversion vs. polymerization time. (b) Mn and Mw/Mn vs. monomer conversion, Figure S14: The calibration curve received for caffeine determination by DPV technique and the relative equation above the curve. The curve depicts linear dependance between the anodic peak current and caffeine concentration in 0.1 M H2SO4 aqueous solution. DP voltammograms received on GCE. DPV parameters: pulse potential of 50 mV, pulse time of 50 ms and scan rate of 50 mV/s, Figure S15: DP voltammograms received on GCE in analysis of coffee samples in 0.1 M H2SO4 aqueous solution using the standard addition method with 1 mL of the sample and after spiking 200, 400 and 600 µL of 10 mM caffeine standard solution, (a) 5% Arabica & Robusta (50/50%) sample; (b) 10% Arabica & Robusta (50/50%) sample. The analysis by standard addition method is presented in the insets. DPV parameters: pulse potential of 50 mV, pulse time of 50 ms and scan rate of 50 mV/s, Figure S16: The calibration curve received for caffeine determination by HPLC technique and the relative equation on the side. Analysis performed with the use of 1290 Infinity LC system with DAD detector and Agilent ZORBAX Eclipse Plus C18, Rapid Resolution HT, 90A, (4.6 x 50 mm, 1.8 um, 600 bar) column in 30°C. Mobile phase: water (75%) and methanol (25%), flow rate: 1.0 ml/min. Wavelength used for caffeine detection: 272 nm, Figure S17. Polymerization of DMAEMA in 7.0 mM caffeine solution: (a) First-order kinetic plot of monomer conversion vs. polymerization time. (b) Mn and Mw/Mn vs. monomer conversion. (c) GPC traces of DMAEMA with DMF + 10 mM LiCl as the eluent. Figure S4: GPC traces of PDMAEMA chains prepared according to (a) Table 1 entry 1; (b) Table 1 entry 2 and (c) Table 1 entry 3 using as eluent DMF + 10 mM LiCl, Figure S5: 1H NMR stack plot of coffee samples: Segafredo Arabica and Segafredo Espresso CASA in D2O (500 MHz, 25°C), Figure S6: GPC traces of PDMAEMA chains prepared according to (a) Table 2 entry 1; (b) Table 2 entry 2 and (c) Table 2 entry 3 using as eluent DMF + 10 mM LiCl, Figure S7: GPC traces of (a) PDMAEMA; (b) PGMA; (c) POEGMA chains prepared according to (a) Table 3 entry 1; (b) Table 3 entry 2 and (c) Table 3 entry 3 using as eluent DMF + 10 mM LiCl, Table S3: ARGET ATRP of n-BA in miniemulsion medium of 10% coffee solution, Figure S8: Polymerization of n-BA in miniemulsion system: (a) First-order kinetic plot of monomer conversion vs. polymerization time. (b) Mn and Mw/Mn vs. monomer conversion. (c) GPC traces of n-BA chains with THF as the eluent, Figure S9: 1H NMR spectrum of PDMAEMA homopolymer (after purification) in CDCl3 (500 MHz, 25°C). The chemical shifts characteristic for PDMAEMA chains were assigned: δ (ppm) = 0.72-1.15 (3H, –CH3, α), 1.72-2.07 (2H, –CH2–, β), 2.23-2.53 (6H, 2 x -CH3, c), 2.57-2.84 (2H, –CH2–, b) and 3.99-4.19 (2H, –CH2–, a), Figure S10: 1H NMR spectrum of PGMA homopolymer (after purification) in CDCl3 (500 MHz, 25°C). The chemical shifts characteristic for PGMA chains were assigned: δ (ppm) = 0.80-1.18 (3H, –CH3, α), 1.84-2.06 (2H, –CH2–, β), 2.56-2.96 (2H, –CH2–, c), 3.15-3.34 (1H, –CH–, b), 3.71-3.89 and 4.19-4.39 (2H, –CH2–, a), Figure S11: 1H NMR spectrum of POEGMA homopolymer (after purification) in CDCl3 (500 MHz, 25°C). The chemical shifts characteristic for POEGMA chains were assigned: δ (ppm) = 0.78-0.97 (3H, –CH3, α), 1.23-1.30 (2H, –CH2–, β), 3.34-3.44 (3H, CH3–, c), 3.54-3.58 (2H, –CH2–, b-first segment), 3.59-3.82 (28H, –CH2– and –CH2–, a + b), and 4.01-4.17 (2H, –CH2–, a-first segment), Figure S12: 1H NMR spectrum of PBA homopolymer (after purification) in CDCl3 (500 MHz, 25°C). The chemical shifts characteristic for PBA chains were assigned: δ (ppm) = 0.85-1.01 (3H, –CH3, d), 1.31-2.00 (6H, –CH2–, a + b + c), 2.20-2.43 (1H, –CH=, α) and 3.90-4.14 (2H, –CH2–, β), Table S4: Chain-extension reaction of PDMAEMA in 7.5% coffee solution, Figure S13: Polymerization of PDMAEMA (first block) in 7.5% coffee solution (a) First-order kinetic plot of monomer conversion vs. polymerization time. (b) Mn and Mw/Mn vs. monomer conversion, Figure S14: Copolymerization of PDMAEMA (second block) in 7.5% coffee solution (a) First-order kinetic plot of monomer conversion vs. polymerization time. (b) Mn and Mw/Mn vs. monomer conversion, Figure S15: The calibration curve received for caffeine determination by DPV technique and the relative equation above the curve. The curve depicts linear dependance between the anodic peak current and caffeine concentration in 0.1 M H2SO4 aqueous solution. DP voltammograms received on GCE. DPV parameters: pulse potential of 50 mV, pulse time of 50 ms and scan rate of 50 mV/s, Figure S16: DP voltammograms received on GCE in analysis of coffee samples in 0.1 M H2SO4 aqueous solution using the standard addition method with 1 mL of the sample and after spiking 200, 400 and 600 µL of 10 mM caffeine standard solution, (a) 5% Arabica & Robusta (50/50%) sample; (b) 10% Arabica & Robusta (50/50%) sample. The analysis by standard addition method is presented in the insets. DPV parameters: pulse potential of 50 mV, pulse time of 50 ms and scan rate of 50 mV/s, Figure S17: The calibration curve received for caffeine determination by HPLC technique and the relative equation on the side. Analysis performed with the use of 1290 Infinity LC system with DAD detector and Agilent ZORBAX Eclipse Plus C18, Rapid Resolution HT, 90A, (4.6 x 50 mm, 1.8 um, 600 bar) column in 30°C. Mobile phase: water (75%) and methanol (25%), flow rate: 1.0 ml/min. Wavelength used for caffeine detection: 272 nm, Table S5. ARGET ATRP of DMAEMA in 7.0 mM caffeine solution, Figure S18: Polymerization of DMAEMA in 7.0 mM caffeine solution: (a) First-order kinetic plot of monomer conversion vs. polymerization time. (b) Mn and Mw/Mn vs. monomer conversion. (c) GPC traces of DMAEMA with DMF + 10 mM LiCl as the eluent. |
12
|
26. Karkare, P.; Kumar, S.; Murthy, C.N. ARGET-ATRP using beta-CD as reducing agent for the synthesis of PMMA-b-PS-b-PMMA triblock copolymers. J. Appl. Polym. Sci. 2019, 136 (9), 47117. 27. Chmielarz, P.; Krys, P.; Park, S.; Matyjaszewski, K. PEO-b-PNIPAM copolymers via SARA ATRP and eATRP in aqueous media. Polymer 2015, 71, 143-147. 28. Fantin, M.; Isse, A.A.; Gennaro, A.; Matyjaszewski, K. Understanding the fundamentals of aqueous ATRP and defining conditions for better control. Macromolecules 2015, 48 (19), 6862-6875. 29. Flejszar, M.; Chmielarz, P.; Smenda, J.; Wolski, K. Following principles of green chemistry: low ppm photo-ATRP of DMAEMA in water/ethanol mixture. Polymer 2021, 228, 123905. 30. Sun, Y.; Zhai, G.Q. CuSO4-catalyzed self-initiated radical polymerization of 2-(N,N-dimethylamino) ethyl methacrylate as an intrinsically reducing inimer. Chinese J. Polym. Sci. 2013, 31 (8), 1161-1172. 31. Chmielarz, P.; Krys, P.; Wang, Z.; Wang, Y.; Matyjaszewski, K. Synthesis of well-defined polymer brushes from silicon wafers via surface-initiated seATRP. Macromol. Chem. Phys. 2017, 218 (11), 1700106. 32. Morits, M.; Hynninen, V.; Nonappa; Niederberger, A.; Ikkala, O.; Gröschel, A.H.; Müllner, M. Polymer brush guided templating on well-defined rod-like cellulose nanocrystals. Polym. Chem. 2018, 9 (13), 1650-1657. 33. Xie, Y.; Chen, S.Q.; Qian, Y.H.; Zhao, W.F.; Zhao, C.S. Photo-responsive membrane surface: switching from bactericidal to bacteria-resistant property. Mater. Sci. Eng. C. 2018, 84, 52-59. 34. Xiong, D.; Zhang, X.F.; Peng, S.Y.; Gu, H.W.; Zhang, L.J. Smart pH-sensitive micelles based on redox degradable polymers as DOX/GNPs carriers for controlled drug release and CT imaging. Colloids Surf. B Biointerfaces 2018, 163, 29-40. 35. Wang, M.; Wang, X.; Zhang, K.; Wu, M.; Wu, Q.; Liu, J.; Yang, J.; Zhang, J. Nano-hydroxyapatite particle brushes via direct initiator tethering and surface-initiated atom transfer radical polymerization for dual responsive pickering emulsion. Langmuir 2020, 36 (5), 1192-1200. 36. Zhang, B.; Yan, Q.; Yuan, S.J.; Zhuang, X.D.; Zhang, F. Enhanced antifouling and anticorrosion properties of stainless steel by biomimetic anchoring PEGDMA-cross-linking polycationic brushes. Ind. Eng. Chem. Res. 2019, 58 (17), 7107-7119. 37. Chmielarz, P.; Park, S.; Simakova, A.; Matyjaszewski, K. Electrochemically mediated ATRP of acrylamides in water. Polymer 2015, 60, 302-307. 38. Michieletto, A.; Lorandi, F.; De Bon, F.; Isse, A.A.; Gennaro, A. Biocompatible polymers via aqueous electrochemically mediated atom transfer radical polymerization. J. Polym. Sci. 2020, 58 (1), 114-123. 39. Ahmed Fawzy, I.A.Z., Khalid S. Khairou, Layla S. Almazroai, Tahani M. Bawazeer, Badriah A. Al-Jahdali. Oxidation of caffeine by permanganate Ion in perchloric and sulfuric acids solutions: a comparative kinetic study. Sci. J. Chem. 2016, 4, 19-28. 40. Burton, I.W.; Farina, C.F.M.; Ragupathy, S.; Arunachalam, T.; Newmaster, S.; Berrue, F. Quantitative NMR methodology for the authentication of roasted coffee and prediction of blends. J. Agric. Food Chem. 2020, 68 (49), 14643-14651. 41. Redivo, L.; Stredansky, M.; De Angelis, E.; Navarini, L.; Resmini, M.; Svorc, L. Bare carbon electrodes as simple and efficient sensors for the quantification of caffeine in commercial beverages. R. Soc. Open Sci. 2018, 5 (5), 172146. 42. Švorc, L.u.; Tomčík, P.; Svítková, J.; Rievaj, M.; Bustin, D. Voltammetric determination of caffeine in beverage samples on bare boron-doped diamond electrode. Food Chem. 2012, 135 (3), 1198-1204. 43. Xia, J.H.; Matyjaszewski, K. Controlled/"living" radical polymerization. Atom transfer radical polymerization using multidentate amine ligands. Macromolecules 1997, 30 (25), 7697-7700. 44. Chmielarz, P.; Sobkowiak, A. Synthesis of poly(butyl acrylate) using an electrochemically mediated atom transfer radical polymerization. Polimery 2016, 61 (9), 585-590. 45. Noein, L.; Haddadi-Asl, V.; Salami-Kalajahi, M. Grafting of pH-sensitive poly (N,N-dimethylaminoethyl methacrylate-co-2-hydroxyethyl methacrylate) onto HNTS via surface-initiated atom transfer radical polymerization for controllable drug release. Int. J. Polym. Mater. Polym. Biomat. 2017, 66 (3), 123-131. 46. Zhang, M.M.; Xiong, Q.Q.; Chen, J.Q.; Wang, Y.S.; Zhang, Q.Q. A novel cyclodextrin-containing pH-responsive star polymer for nanostructure fabrication and drug delivery. Polym. Chem. 2013, 4 (19), 5086-5095. 47. Ding, A.S.; Xu, J.; Gu, G.X.; Lu, G.L.; Huang, X.Y. PHEA-g-PMMA well-defined graft copolymer: ATRP synthesis, self-assembly, and synchronous encapsulation of both hydrophobic and hydrophilic guest molecules. Sci Rep 2017, 7, 12601.
|
26. Karkare, P.; Kumar, S.; Murthy, C.N. ARGET-ATRP using beta-CD as reducing agent for the synthesis of PMMA-b-PS-b-PMMA triblock copolymers. J. Appl. Polym. Sci. 2019, 136 (9), 47117. 27. Wei, Q.; Sun, M.; Lorandi, F.; Yin, R.; Yan, J.; Liu, T.; Kowalewski, T.; Matyjaszewski, K. Cu-catalyzed atom transfer radical polymerization in the presence of liquid metal micro/nanodroplets. Macromolecules 2021, 54 (4), 1631-1638. 28. Chmielarz, P.; Krys, P.; Park, S.; Matyjaszewski, K. PEO-b-PNIPAM copolymers via SARA ATRP and eATRP in aqueous media. Polymer 2015, 71, 143-147. 29. Fantin, M.; Isse, A.A.; Gennaro, A.; Matyjaszewski, K. Understanding the fundamentals of aqueous ATRP and defining conditions for better control. Macromolecules 2015, 48 (19), 6862-6875. 30. Flejszar, M.; Chmielarz, P.; Smenda, J.; Wolski, K. Following principles of green chemistry: low ppm photo-ATRP of DMAEMA in water/ethanol mixture. Polymer 2021, 228, 123905. 31. Sun, Y.; Zhai, G.Q. CuSO4-catalyzed self-initiated radical polymerization of 2-(N,N-dimethylamino) ethyl methacrylate as an intrinsically reducing inimer. Chinese J. Polym. Sci. 2013, 31 (8), 1161-1172. 32. Chmielarz, P.; Krys, P.; Wang, Z.; Wang, Y.; Matyjaszewski, K. Synthesis of well-defined polymer brushes from silicon wafers via surface-initiated seATRP. Macromol. Chem. Phys. 2017, 218 (11), 1700106. 33. Morits, M.; Hynninen, V.; Nonappa; Niederberger, A.; Ikkala, O.; Gröschel, A.H.; Müllner, M. Polymer brush guided templating on well-defined rod-like cellulose nanocrystals. Polym. Chem. 2018, 9 (13), 1650-1657. 34. Xie, Y.; Chen, S.Q.; Qian, Y.H.; Zhao, W.F.; Zhao, C.S. Photo-responsive membrane surface: switching from bactericidal to bacteria-resistant property. Mater. Sci. Eng. C. 2018, 84, 52-59. 35. Xiong, D.; Zhang, X.F.; Peng, S.Y.; Gu, H.W.; Zhang, L.J. Smart pH-sensitive micelles based on redox degradable polymers as DOX/GNPs carriers for controlled drug release and CT imaging. Colloids Surf. B Biointerfaces 2018, 163, 29-40. 36. Wang, M.; Wang, X.; Zhang, K.; Wu, M.; Wu, Q.; Liu, J.; Yang, J.; Zhang, J. Nano-hydroxyapatite particle brushes via direct initiator tethering and surface-initiated atom transfer radical polymerization for dual responsive pickering emulsion. Langmuir 2020, 36 (5), 1192-1200. 37. Zhang, B.; Yan, Q.; Yuan, S.J.; Zhuang, X.D.; Zhang, F. Enhanced antifouling and anticorrosion properties of stainless steel by biomimetic anchoring PEGDMA-cross-linking polycationic brushes. Ind. Eng. Chem. Res. 2019, 58 (17), 7107-7119. 38. Chmielarz, P.; Park, S.; Simakova, A.; Matyjaszewski, K. Electrochemically mediated ATRP of acrylamides in water. Polymer 2015, 60, 302-307. 39. Michieletto, A.; Lorandi, F.; De Bon, F.; Isse, A.A.; Gennaro, A. Biocompatible polymers via aqueous electrochemically mediated atom transfer radical polymerization. J. Polym. Sci. 2020, 58 (1), 114-123. 40. Ahmed Fawzy, I.A.Z., Khalid S. Khairou, Layla S. Almazroai, Tahani M. Bawazeer, Badriah A. Al-Jahdali. Oxidation of caffeine by permanganate Ion in perchloric and sulfuric acids solutions: a comparative kinetic study. Sci. J. Chem. 2016, 4, 19-28. 41. Krys, P.; Matyjaszewski, K. Kinetics of atom transfer radical polymerization. Eur. Polym. J. 2017, 89, 482-523. 42. Fantin, M.; Chmielarz, P.; Wang, Y.; Lorandi, F.; Isse, A.A.; Gennaro, A.; Matyjaszewski, K. Harnessing the interaction between surfactant and hydrophilic catalyst to control eATRP in miniemulsion. Macromolecules 2017, 50 (9), 3726-3732. 43. Ribelli, T.G.; Lorandi, F.; Fantin, M.; Matyjaszewski, K. Atom transfer radical polymerization: billion times more active catalysts and new initiation systems. Macromol. Rapid Commun. 2019, 40 (1), 1800616. 44. Chmielarz, P.; Sobkowiak, A.; Matyjaszewski, K. A simplified electrochemically mediated ATRP synthesis of PEO-b-PMMA copolymers. Polymer 2015, 77, 266-271. 45. Burton, I.W.; Farina, C.F.M.; Ragupathy, S.; Arunachalam, T.; Newmaster, S.; Berrue, F. Quantitative NMR methodology for the authentication of roasted coffee and prediction of blends. J. Agric. Food Chem. 2020, 68 (49), 14643-14651. 46. Švorc, L.u.; Tomčík, P.; Svítková, J.; Rievaj, M.; Bustin, D. Voltammetric determination of caffeine in beverage samples on bare boron-doped diamond electrode. Food Chem. 2012, 135 (3), 1198-1204. 47. Redivo, L.; Stredansky, M.; De Angelis, E.; Navarini, L.; Resmini, M.; Svorc, L. Bare carbon electrodes as simple and efficient sensors for the quantification of caffeine in commercial beverages. R. Soc. Open Sci. 2018, 5 (5), 172146. 48. Xia, J.H.; Matyjaszewski, K. Controlled/"living" radical polymerization. Atom transfer radical polymerization using multidentate amine ligands. Macromolecules 1997, 30 (25), 7697-7700. 49. Chmielarz, P.; Sobkowiak, A. Synthesis of poly(butyl acrylate) using an electrochemically mediated atom transfer radical polymerization. Polimery 2016, 61 (9), 585-590. 50. Noein, L.; Haddadi-Asl, V.; Salami-Kalajahi, M. Grafting of pH-sensitive poly (N,N-dimethylaminoethyl methacrylate-co-2-hydroxyethyl methacrylate) onto HNTS via surface-initiated atom transfer radical polymerization for controllable drug release. Int. J. Polym. Mater. Polym. Biomat. 2017, 66 (3), 123-131. 51. Zhang, M.M.; Xiong, Q.Q.; Chen, J.Q.; Wang, Y.S.; Zhang, Q.Q. A novel cyclodextrin-containing pH-responsive star polymer for nanostructure fabrication and drug delivery. Polym. Chem. 2013, 4 (19), 5086-5095. 52. Ding, A.S.; Xu, J.; Gu, G.X.; Lu, G.L.; Huang, X.Y. PHEA-g-PMMA well-defined graft copolymer: ATRP synthesis, self-assembly, and synchronous encapsulation of both hydrophobic and hydrophilic guest molecules. Sci Rep 2017, 7, 12601. |
15-16 |

Reviewer 2 Report
This manuscript described an interesting new method of performing ATRP in aqueous systems, coffee solution, in particular. The reducing nature of caffeine present in coffee generates the ATRP activator CuI/L from CuII/L, driving the polymerization forward. The results and discussions are mostly clear, but the reviewer found several parts of the discussion that should be improved:
- It is generally agreed that Cu-Br bond could easily dissociate in aqueous systems, which could cause high dispersity. And adding bromide salts (e.g., NaBr) could improve the control. So, the authors could try 1-2 polymerizations with the addition of bromides to see if the dispersity could be decreased, especially for the DMAEMA system.
- Lines 52-53: The reviewer suggests that the authors could cite some additional recent systems that use novel external reducing agents, such as liquid metal (Macromolecules 2021, 54, 1631–1638)
- Line 116: ATP control, should be ATRP control.
- Table 1, initiation efficiency: the initiation efficiency was calculated based on the apparent molecular weight, which was determined by PS standard from GPC. However, this should be different from the absolute MW, due to the different refractive indices between PS and PDMAEMA. The authors should comment on the possible errors caused by this type of MW calculation.
- The reviewer also suggests 1-2 additional polymerization experiments with pure caffeine aqueous solution.
Author Response
Manuscript number: molecules-1529687
Coffee beverage: A new strategy for the synthesis of polymethacrylates via ATRP
We thank the reviewers for their valuable comments and suggestions. All changes and amendments are highlighted in the new text. Our point-by-point response follows:
Response to Reviewer 2:
Comment 1: It is generally agreed that Cu-Br bond could easily dissociate in aqueous systems, which could cause high dispersity. And adding bromide salts (e.g., NaBr) could improve the control. So, the authors could try 1-2 polymerizations with the addition of bromides to see if the dispersity could be decreased, especially for the DMAEMA system.
We agree with you, the X-CuII/L bond can easily dissociate in aqueous media, typically high concentration of catalyst is needed or the presence of salts with halide anions is required. Because of experiments were carried out in low ppm concentration of catalyst (50 ppm), each reaction contained halide salt NaBr in concentration 0.1 M (with respect to the total volume) as presented in Supplementary Material at section S1 „Synthesis procedures”. In order to dispel all doubts, we added NaBr concentration to each table footer. As suggested, we performed additional experiments in coffee extract solution (Arabica & Robusta 7.5%) without addition of halide salt as well as in solution contained 0.3 M NaBr. Indeed, the lack of a halide salt caused a loss of process control due to ineffective catalyst regeneration, the addition of NaBr resulted in an increase of control over polymerization reflected by a decrease in initiation efficiency and a narrower molecular weight distribution. A higher salt concentration affected more effective catalyst regeneration and an increase in control over the process (compare initiation efficiency), however, the reaction stopped after 0.5 hour did not reach a satisfactory MWD.
The table and graphs presented below have been included in the Supplementary Material in chapter S3 “ARGET ATRP in various NaBr concentration”
Table 1. ARGET ATRP in various NaBr concentration in 7.5% solution received from equally proportional blend of Arabica and Robusta beans.
Entry1 |
NaBr conc.2 (M) |
Conv3 (%) |
DPapp3 |
kpapp3 (h-1) |
Mn,th4 (x10-3) |
Mn,app5 (x10-3) |
Ð5 |
f6 (%) |
1 |
0.0 |
29.5 |
59 |
0.70 |
9.5 |
78.0 |
2.04 |
819 |
2 |
0.1 |
70.9 |
142 |
2.55 |
22.6 |
62.8 |
1.63 |
278 |
3 |
0.3 |
58.0 |
116 |
1.71 |
18.5 |
41.3 |
1.92 |
223 |
1 General reaction conditions: [DMAEMA]0/[EBiB]0/[CuIIBr2]0/[TPMA]0: 200/1/0.01/0.02; T = 22°C; Vtot = 8 mL (DMAEMA/solvent = 0.4/0.6 by v/v), [DMAEMA]0 = 19 mM, [I]0 = 0.09 mM. [CuIIBr2]0 = 1.19 µM (2-fold excess of TPMA), t = 0.5 h. Solvent: 7.5% solution of Arabica & Robusta (50/50%).
2 Concentration with respect to the total volume.
3 Monomer conversion, polymerization rate (kpapp) and DPapp were determined by 1H NMR.
4 Mn,th = ([M]0/[I]0) × Conv × Mmonomer + Minitiator.
5 Mn,app and Ð was determined by DMF + 10 mM LiCl GPC with PS standards.
6 Initiation efficiency calculated as: f = Mn,app/Mn,th.
Figures attached in pdf file
(a) (b)
Figure 1. (a) Semilogarithmic kinetic plot, (b) Mn and Mw/Mn vs. monomer conversion for ARGET ATRP in preparation of PDMAEMA in various NaBr concentration (Table 1). Reactions conditions as in Table 1.
An additional comment was added in the Results of the Main text (page 4, highlighted version).
Comment 2: Lines 52-53: The reviewer suggests that the authors could cite some additional recent systems that use novel external reducing agents, such as liquid metal (Macromolecules 2021, 54, 1631–1638)
As suggested, we mentioned also about novel external reducing agents as liquid metal in regard to the efficient reducing compounds at page 2. (Macromolecules 2021, 54, 1631–1638).
An additional information was added in the Introduction of the Main text (page 2, highlighted version).
Comment 3: Line 116: ATP control, should be ATRP control.
Thank for your attention, we corrected ATP to ATRP
The correction was done in the Results of the Main text (page 4, highlighted version).
Comment 4: The initiation efficiency was calculated based on the apparent molecular weight, which was determined by PS standard from GPC. However, this should be different from the absolute MW, due to the different refractive indices between PS and PDMAEMA. The authors should comment on the possible errors caused by this type of MW calculation.
As mentioned, the initiation efficiency was calculated based on the apparent molecular weight, which was determined by PS standard from GPC. However, molecular weight was determined in relation to internal standard - toluene. The GPC columns were calibrated by PS standard but the molecular weight of obtained polymer is determined in relation to toluene mass, which is added to each measured samples. Thanks to it, the possible errors in measuring absolute MW could be minimized.
Indeed, the refractive index indices do differ; the refractive index value of polystyrene is 1.592 [1], while that of acrylic polymer is 1.491 [1], therefore a slight error will occur. However, it will be minimal because we are considering linear systems, not branched systems, for which the Multi Angle Light Scattering (MALS) detector should be used [2].
- Sultanova, N.; Kasarova, S.; Nikolov, I. Dispersion properties of optical polymers. Acta Physica Polonica A 2009, 116 (4), 585-587.
- Principles of detection and characterization of branching in synthetic and natural polymers by MALS. The Column 2014, 10 (10), 1-35.
Comment 5: The reviewer also suggests 1-2 additional polymerization experiments with pure caffeine aqueous solution.
Thank for your suggestion. We have performed such polymerization previously and the detailed results are in Supplementary Materials at section S10 "ARGET ATRP of DMAEMA in 7.0 mM caffeine solution". In the main manuscript we also mentioned this reaction at page 9. Obtained the low dispersity polymer product in the coffee beverages as well as in pure caffeine solution. In addition, the polymerization rate is influenced by the strength of the coffee brew - the blend of Arabica and Robusta beans showed stronger antioxidant properties compared to the same amount of pure Arabica, which increased the polymerization rate. The weakest antioxidant effect was observed in a pure caffeine solution that does not contain other antioxidant compounds. In this case, the polymerization rate and monomer conversion are the lowest.
Additionally, English language and style was checked carefully. All corrections have been made as follows:
Before correction |
Corrected version in revised text |
Page of the revised text (highlighted version) |
Coffee − the most popular beverage in the 21st-century society … |
Coffee − the most popular beverage in the 21st- 21st century society |
1 |
|
|
|
pure Arabica beans and proportional blend of Arabica and Robusta beans |
pure Arabica beans and a proportional blend of Arabica and Robusta beans |
1 |
…significant antioxidant capacity |
…significant antioxidant activitycapacity |
1 |
The highest amount of compounds with antioxidant potential was proved in medium roast coffee due to..… |
The highest amount of compounds with antioxidant potential was determinedproved in medium roast coffee grains due to… |
1 |
There are over five hundred types of coffee |
There are over five hundred typesspecies of coffee |
1 |
So far, mainly vitamins such as ascorbic acid [20-23] or riboflavin [12, 24] and sugars such as glucose [25] and cyclodextrin [26]. |
So far, mainly used vitamins such as ascorbic acid [20-23] or riboflavin [12, 24] and sugars such as glucose [25] and cyclodextrin [26], but also the novel external reducing agents as liquid metal [27] were used. |
2 |
Equally important is the solubility of the monomers in the individual media, which affects the choice of the reaction solvent. |
Equally important is the solubility of the monomers in the individual media, which determines the selectionaffects the choice of the reaction solvent. |
2 |
Antioxidants derived from a coffee extract as caffeine, chlorogenic acids, cafestol, kahweol and trigonelline trigger a reduction of a deactivator complex, X-CuII/L, ensuring continuous catalytic reaction. |
Antioxidants derived from a coffee extract such as caffeine, chlorogenic acids, cafestol, kahweol and trigonelline trigger a reduction of a deactivator complex, X-CuII/L, ensuring continuous catalytic reaction. Voltammetric measurements were carried out, which confirmed the strongest antioxidant effect of caffeine in both mixtures, therefore the mechanism in which caffeine is the dominant antioxidant factor controlling the polymerization process was presented. |
3 |
Caffeine, being the main antioxidant of coffee,… |
Caffeine, being the main antioxidant present inof coffee,… |
3 |
The process takes place in a mild oxidizing environment of copper (II) bromide, … |
The process takes place in a mild oxidizing environmentmedium of copper(II) bromide, … |
3 |
The mechanism involves the attack of the oxidant - deactivator on the caffeine molecule to give a free radical derived from caffeine. |
The mechanism involves the attackinteraction of the oxidant - deactivator onwith the caffeine molecule to give a free radical derived from caffeine. |
3 |
Then, water particles from the reaction solvent react with a free radical, giving 8-hydroxy caffeine which fast tautemerise to the final oxidation product of caffeine - 1,3,7-trimethyluric acid. [40] |
Then, water particles from the reaction solvent react with a free radical, giving 8-hydroxy caffeine which fast tautemerisetautomerise to the final oxidation product of caffeine - 1,3,7-trimethyluric acid (Figure 1) [40]. According to the ATRP mechanism, the dynamic equilibrium of the copper(I) and copper(II) complex is involved in controlling polymerization. The complex of copper(II) introduced into the polymerization system undergoes reduction due to the caf-feine action. Additionally, trace amounts of oxygen could cause oxidation of the copper and affect the dynamic equilibrium of CuII to CuI estabilishing. The polymerization process is then controlled by appropriate ratio of activator to deactivator species [8, 23]. |
3 |
Therefore, all next polymerization processes were performed at room temperature, allowing to achieve products in controlled manner and in a shorter time. For the synthesis in pure water without the addition of a reducing agent |
Therefore, all next polymerization processes were performed at room temperature, al-lowing to achieve products in controlled manner and in a shorter time. In efficient ATRP catalyst system, metal center must have two easily accessible oxidation states as well as must be able to accept the halogen, which requires a suffi-ciently high halogenophilicity [41]. Low halidophilicity i.e. the association of the halide an-ion to the catalyst in the higher oxidation state, related to the connection of the halo-gen ion create possibility of disproportionation of CuI/L to X-CuII/L and Cu0 [42, 43]. Finally, halide anions must have high affinity for CuIIL2+, such as to provide the pres-ence of enough concentration of deactivator to give a well-controlled process. In aqueous systems, the X-CuII/L bond can easily dissociate to CuIIL2+ and Br−, typically required is high concentration of catalyst is needed or the presence of salts with halide anions in order to shift the equilibrium toward the X-CuII/L deactivator species [28, 44]. Because of experiments were carried out in low ppm concentration of catalyst (9.5 ppm by wt%), the effect of the addition of a halide salt on the polymerization process in the medium of a coffee blend extract was investigated (Arabica & Robusta 1:1). Indeed, the lack of a halide salt caused a loss of process control due to ineffective catalyst regeneration, the addition of NaBr resulted in an increase of control over polymerization reflected valid initiation efficiency values and a narrower molecular weight distribution (Table S2). Higher salt concentration affected more effective catalyst re-generation and an increase in control over the process (compare initiation efficiency in Table S2), however, the reaction stopped after 0.5 hour did not reach a satisfactory MWD (Figure S3b), therefore it was decided to conduct subsequent reactions at a sodium bromide concentration equal to 0.1 M. For the synthesis in pure water without the addition of a reducing agent |
4 |
On the other hand, the N-CH2 group remains the initiation site of radical polymerization, resulting in a loss of ATP control as indicated by an almost 900% initiation efficiency. |
On the other hand, the N-CH2 group remains the initiation site of radical polymerization, resulting in a loss of ATPATRP control as indicated by an almost 900% initiation efficiency. |
4 |
Table 1. footer: 1 General reaction conditions: [DMAEMA]0/[EBiB]0/[CuIIBr2]0/[TPMA]0: 200/1/0.01/0.02; T = 22°C; Vtot = 8 mL (DMAEMA/solvent = 0.4/0.6 by v/v), [DMAEMA]0 = 19 mM, [I]0 = 0.09 mM. [CuIIBr2]0 = 1.19 µM (2-fold excess of TPMA), t = 0.5 h, except entry 1: t = 2h.
|
Table 1. footer: 1 General reaction conditions: [DMAEMA]0/[EBiB]0/[CuIIBr2]0/[TPMA]0: 200/1/0.01/0.02; T = 22°C; Vtot = 8 mL (DMAEMA/solvent = 0.4/0.6 by v/v), [DMAEMA]0 = 19 mM, [I]0 = 0.09 mM. [NaBr] = 0.1 M, [CuIIBr2]0 = 1.19 µM (2-fold excess of TPMA), t = 0.5 h, except entry 1: t = 2h. |
5 |
caused by low amount of catalytic complex – 9.5 ppm by wt%, obtained PDMAEMA |
caused by low amount of catalytic complex – 9.5 ppm by wt%, obtained PDMAEMA |
5 |
Both types of solutions received from different blends of coffee grains were analyzed by 1H NMR (Figure S4) to confirm |
Both types of solutions received from different blends of coffee grains were analyzed by 1H NMR (Figure S4Figure S5) to confirm |
5 |
Table 2. footer: 1 General reaction conditions: [DMAEMA]0/[EBiB]0/[CuIIBr2]0/[TPMA]0: 200/1/0.01/0.02; T = 22°C; Vtot = 8 mL (DMAEMA/solvent = 0.4/0.6 by v/v), [DMAEMA]0 = 19 mM, [I]0 = 0.09 mM. [CuIIBr2]0 = 1.19 µM (2-fold excess of TPMA), t = 0.5 h. |
Table 2. footer: 1 General reaction conditions: [DMAEMA]0/[EBiB]0/[CuIIBr2]0/[TPMA]0: 200/1/0.01/0.02; T = 22°C; Vtot = 8 mL (DMAEMA/solvent = 0.4/0.6 by v/v), [DMAEMA]0 = 19 mM, [I]0 = 0.09 mM. [NaBr] = 0.1 M, [CuIIBr2]0 = 1.19 µM (2-fold excess of TPMA), t = 0.5 h. |
6 |
Table 3. footer: 1General reaction conditions: [Monomer]0/[EBiB]0: 200/1; [CuIIBr2]= 9.5 ppm by wt (2-fold excess of TPMA); T = 22°C; Vtot = 8 mL (monomer/solvent = 0.4/0.6 by v/v). |
Table 3. footer: 1General reaction conditions: [Monomer]0/[EBiB]0: 200/1; [CuIIBr2]= 9.5 ppm by wt (2-fold excess of TPMA); [NaBr] = 0.1 M, T = 22°C; Vtot = 8 mL (monomer/solvent = 0.4/0.6 by v/v). |
7 |
In order to investigate the chain-end fidelity, chain extension experiment of PDMAEMA were performed. |
In order to investigate the chain-end fidelity, chain extension experiment of PDMAEMA werewas performed. |
8 |
The PDMAEMA constituting the first block of the copolymers was characterized by monomodal GPC trace which points to the high retention of chain-end functionality (Figure 5, Figure S12). |
The PDMAEMA constituting the first block of the copolymers was characterized by monomodal GPC trace which points to the high retention of chain-end functionality (Figure 5, Figure S1213). |
8 |
The second block was successfully incorporated in situ in a controlled manner proved by linear first-order kinetics (Figure S13) and a shift in the MW peak toward to higher molecular mass (Figure 5). |
The second block was successfully incorporated in situ in a controlled manner proved by linear first-order kinetics (Figure S1314) and a shift in the MW peak toward to higher molecular mass (Figure 5). |
8 |
Three independent measurements were made for the all points included on the graphs (insets in the Figure 7 and Figure S15). |
Three independent measurements were made for the all points included on the graphs (insets in the Figure 7 and Figure S1516). |
9 |
The caffeine was successfully used as an electron donor in the ARGET ATRP technique, which also confirms reaction carried out in 7.0 mM pure caffeine solution (Table S4). |
The caffeine was successfully used as an electron donor in the ARGET ATRP technique, which also confirms reaction carried out in 7.0 mM pure caffeine solution (Table S4Table S5). |
10 |
The calibration curve equation is reported in point S.8. |
The calibration curve equation is reported in point S.8 in Supplementary Materials. |
12 |
Table S1: ARGET ATRP of DMAEMA in various temperature, Figure S1: (a) Semilogarithmic kinetic plot, (b) Mn and Mw/Mn vs. monomer conversion for preparation of PDMAEMA in coffee solution at various temperature (Table S1, entries 1-2). Reaction conditions as in Table S1, Figure S2: GPC traces of PDMAEMA chains prepared according to (a) Table S1 entry 1 and (b) Table S1 entry 2 using as eluent DMF + 10 mM LiCl, Figure S3: GPC traces of PDMAEMA chains prepared according to (a) Table 1 entry 1; (b) Table 1 entry 2 and (c) Table 1 entry 3 using as eluent DMF + 10 mM LiCl, Figure S4: 1H NMR stack plot of coffee samples: Segafredo Arabica and Segafredo Espresso CASA in D2O (500 MHz, 25°C), Figure S5: GPC traces of PDMAEMA chains prepared according to (a) Table 2 entry 1; (b) Table 2 entry 2 and (c) Table 2 entry 3 using as eluent DMF + 10 mM LiCl, Figure S6: GPC traces of (a) PDMAEMA; (b) PGMA; (c) POEGMA chains prepared according to (a) Table 3 entry 1; (b) Table 3 entry 2 and (c) Table 3 entry 3 using as eluent DMF + 10 mM LiCl, Table S2: ARGET ATRP of n-BA in miniemulsion medium of 10% coffee solution, Figure S7: Polymerization of n-BA in miniemulsion system: (a) First-order kinetic plot of monomer conversion vs. polymerization time. (b) Mn and Mw/Mn vs. monomer conversion. (c) GPC traces of n-BA chains with THF as the eluent, Figure S8: 1H NMR spectrum of PDMAEMA homopolymer (after purification) in CDCl3 (500 MHz, 25°C). The chemical shifts characteristic for PDMAEMA chains were assigned: δ (ppm) = 0.72-1.15 (3H, –CH3, α), 1.72-2.07 (2H, –CH2–, β), 2.23-2.53 (6H, 2 x -CH3, c), 2.57-2.84 (2H, –CH2–, b) and 3.99-4.19 (2H, –CH2–, a), Figure S9: 1H NMR spectrum of PGMA homopolymer (after purification) in CDCl3 (500 MHz, 25°C). The chemical shifts characteristic for PGMA chains were assigned: δ (ppm) = 0.80-1.18 (3H, –CH3, α), 1.84-2.06 (2H, –CH2–, β), 2.56-2.96 (2H, –CH2–, c), 3.15-3.34 (1H, –CH–, b), 3.71-3.89 and 4.19-4.39 (2H, –CH2–, a), Figure S10: 1H NMR spectrum of POEGMA homopolymer (after purification) in CDCl3 (500 MHz, 25°C). The chemical shifts characteristic for POEGMA chains were assigned: δ (ppm) = 0.78-0.97 (3H, –CH3, α), 1.23-1.30 (2H, –CH2–, β), 3.34-3.44 (3H, CH3–, c), 3.54-3.58 (2H, –CH2–, b-first segment), 3.59-3.82 (28H, –CH2– and –CH2–, a + b), and 4.01-4.17 (2H, –CH2–, a-first segment), Figure S11: 1H NMR spectrum of PBA homopolymer (after purification) in CDCl3 (500 MHz, 25°C). The chemical shifts characteristic for PBA chains were assigned: δ (ppm) = 0.85-1.01 (3H, –CH3, d), 1.31-2.00 (6H, –CH2–, a + b + c), 2.20-2.43 (1H, –CH=, α) and 3.90-4.14 (2H, –CH2–, β), Table S3: Chain-extension reaction of PDMAEMA in 7.5% coffee solution, Figure S12: Polymerization of PDMAEMA (first block) in 7.5% coffee solution (a) First-order kinetic plot of monomer conversion vs. polymerization time. (b) Mn and Mw/Mn vs. monomer conversion, Figure S13: Copolymerization of PDMAEMA (second block) in 7.5% coffee solution (a) First-order kinetic plot of monomer conversion vs. polymerization time. (b) Mn and Mw/Mn vs. monomer conversion, Figure S14: The calibration curve received for caffeine determination by DPV technique and the relative equation above the curve. The curve depicts linear dependance between the anodic peak current and caffeine concentration in 0.1 M H2SO4 aqueous solution. DP voltammograms received on GCE. DPV parameters: pulse potential of 50 mV, pulse time of 50 ms and scan rate of 50 mV/s, Figure S15: DP voltammograms received on GCE in analysis of coffee samples in 0.1 M H2SO4 aqueous solution using the standard addition method with 1 mL of the sample and after spiking 200, 400 and 600 µL of 10 mM caffeine standard solution, (a) 5% Arabica & Robusta (50/50%) sample; (b) 10% Arabica & Robusta (50/50%) sample. The analysis by standard addition method is presented in the insets. DPV parameters: pulse potential of 50 mV, pulse time of 50 ms and scan rate of 50 mV/s, Figure S16: The calibration curve received for caffeine determination by HPLC technique and the relative equation on the side. Analysis performed with the use of 1290 Infinity LC system with DAD detector and Agilent ZORBAX Eclipse Plus C18, Rapid Resolution HT, 90A, (4.6 x 50 mm, 1.8 um, 600 bar) column in 30°C. Mobile phase: water (75%) and methanol (25%), flow rate: 1.0 ml/min. Wavelength used for caffeine detection: 272 nm, Figure S17. Polymerization of DMAEMA in 7.0 mM caffeine solution: (a) First-order kinetic plot of monomer conversion vs. polymerization time. (b) Mn and Mw/Mn vs. monomer conversion. (c) GPC traces of DMAEMA with DMF + 10 mM LiCl as the eluent. |
Table S1: ARGET ATRP of DMAEMA in various temperature, Figure S1: (a) Semilogarithmic kinetic plot, (b) Mn and Mw/Mn vs. monomer conversion for preparation of PDMAEMA in coffee solution at various temperature (Table S1, entries 1-2). Reaction conditions as in Table S1, Figure S2: GPC traces of PDMAEMA chains prepared according to (a) Table S1 entry 1 and (b) Table S1 entry 2 using as eluent DMF + 10 mM LiCl, Table S2. ARGET ATRP in various NaBr concentration in 7.5% solution received from equally proportional blend of Arabica and Robusta beans, Figure S3. (a) Semilogarithmic kinetic plot, (b) Mn and Mw/Mn vs. monomer conversion for ARGET ATRP in preparation of PDMAEMA in various NaBr concentration (Table S2). Reactions conditions as in Table S2. Figure S3: GPC traces of PDMAEMA chains prepared according to (a) Table 1 entry 1; (b) Table 1 entry 2 and (c) Table 1 entry 3 using as eluent DMF + 10 mM LiCl, Figure S4: 1H NMR stack plot of coffee samples: Segafredo Arabica and Segafredo Espresso CASA in D2O (500 MHz, 25°C), Figure S5: GPC traces of PDMAEMA chains prepared according to (a) Table 2 entry 1; (b) Table 2 entry 2 and (c) Table 2 entry 3 using as eluent DMF + 10 mM LiCl, Figure S6: GPC traces of (a) PDMAEMA; (b) PGMA; (c) POEGMA chains prepared according to (a) Table 3 entry 1; (b) Table 3 entry 2 and (c) Table 3 entry 3 using as eluent DMF + 10 mM LiCl, Table S2: ARGET ATRP of n-BA in miniemulsion medium of 10% coffee solution, Figure S7: Polymerization of n-BA in miniemulsion system: (a) First-order kinetic plot of monomer conversion vs. polymerization time. (b) Mn and Mw/Mn vs. monomer conversion. (c) GPC traces of n-BA chains with THF as the eluent, Figure S8: 1H NMR spectrum of PDMAEMA homopolymer (after purification) in CDCl3 (500 MHz, 25°C). The chemical shifts characteristic for PDMAEMA chains were assigned: δ (ppm) = 0.72-1.15 (3H, –CH3, α), 1.72-2.07 (2H, –CH2–, β), 2.23-2.53 (6H, 2 x -CH3, c), 2.57-2.84 (2H, –CH2–, b) and 3.99-4.19 (2H, –CH2–, a), Figure S9: 1H NMR spectrum of PGMA homopolymer (after purification) in CDCl3 (500 MHz, 25°C). The chemical shifts characteristic for PGMA chains were assigned: δ (ppm) = 0.80-1.18 (3H, –CH3, α), 1.84-2.06 (2H, –CH2–, β), 2.56-2.96 (2H, –CH2–, c), 3.15-3.34 (1H, –CH–, b), 3.71-3.89 and 4.19-4.39 (2H, –CH2–, a), Figure S10: 1H NMR spectrum of POEGMA homopolymer (after purification) in CDCl3 (500 MHz, 25°C). The chemical shifts characteristic for POEGMA chains were assigned: δ (ppm) = 0.78-0.97 (3H, –CH3, α), 1.23-1.30 (2H, –CH2–, β), 3.34-3.44 (3H, CH3–, c), 3.54-3.58 (2H, –CH2–, b-first segment), 3.59-3.82 (28H, –CH2– and –CH2–, a + b), and 4.01-4.17 (2H, –CH2–, a-first segment), Figure S11: 1H NMR spectrum of PBA homopolymer (after purification) in CDCl3 (500 MHz, 25°C). The chemical shifts characteristic for PBA chains were assigned: δ (ppm) = 0.85-1.01 (3H, –CH3, d), 1.31-2.00 (6H, –CH2–, a + b + c), 2.20-2.43 (1H, –CH=, α) and 3.90-4.14 (2H, –CH2–, β), Table S3: Chain-extension reaction of PDMAEMA in 7.5% coffee solution, Figure S12: Polymerization of PDMAEMA (first block) in 7.5% coffee solution (a) First-order kinetic plot of monomer conversion vs. polymerization time. (b) Mn and Mw/Mn vs. monomer conversion, Figure S13: Copolymerization of PDMAEMA (second block) in 7.5% coffee solution (a) First-order kinetic plot of monomer conversion vs. polymerization time. (b) Mn and Mw/Mn vs. monomer conversion, Figure S14: The calibration curve received for caffeine determination by DPV technique and the relative equation above the curve. The curve depicts linear dependance between the anodic peak current and caffeine concentration in 0.1 M H2SO4 aqueous solution. DP voltammograms received on GCE. DPV parameters: pulse potential of 50 mV, pulse time of 50 ms and scan rate of 50 mV/s, Figure S15: DP voltammograms received on GCE in analysis of coffee samples in 0.1 M H2SO4 aqueous solution using the standard addition method with 1 mL of the sample and after spiking 200, 400 and 600 µL of 10 mM caffeine standard solution, (a) 5% Arabica & Robusta (50/50%) sample; (b) 10% Arabica & Robusta (50/50%) sample. The analysis by standard addition method is presented in the insets. DPV parameters: pulse potential of 50 mV, pulse time of 50 ms and scan rate of 50 mV/s, Figure S16: The calibration curve received for caffeine determination by HPLC technique and the relative equation on the side. Analysis performed with the use of 1290 Infinity LC system with DAD detector and Agilent ZORBAX Eclipse Plus C18, Rapid Resolution HT, 90A, (4.6 x 50 mm, 1.8 um, 600 bar) column in 30°C. Mobile phase: water (75%) and methanol (25%), flow rate: 1.0 ml/min. Wavelength used for caffeine detection: 272 nm, Figure S17. Polymerization of DMAEMA in 7.0 mM caffeine solution: (a) First-order kinetic plot of monomer conversion vs. polymerization time. (b) Mn and Mw/Mn vs. monomer conversion. (c) GPC traces of DMAEMA with DMF + 10 mM LiCl as the eluent. Figure S4: GPC traces of PDMAEMA chains prepared according to (a) Table 1 entry 1; (b) Table 1 entry 2 and (c) Table 1 entry 3 using as eluent DMF + 10 mM LiCl, Figure S5: 1H NMR stack plot of coffee samples: Segafredo Arabica and Segafredo Espresso CASA in D2O (500 MHz, 25°C), Figure S6: GPC traces of PDMAEMA chains prepared according to (a) Table 2 entry 1; (b) Table 2 entry 2 and (c) Table 2 entry 3 using as eluent DMF + 10 mM LiCl, Figure S7: GPC traces of (a) PDMAEMA; (b) PGMA; (c) POEGMA chains prepared according to (a) Table 3 entry 1; (b) Table 3 entry 2 and (c) Table 3 entry 3 using as eluent DMF + 10 mM LiCl, Table S3: ARGET ATRP of n-BA in miniemulsion medium of 10% coffee solution, Figure S8: Polymerization of n-BA in miniemulsion system: (a) First-order kinetic plot of monomer conversion vs. polymerization time. (b) Mn and Mw/Mn vs. monomer conversion. (c) GPC traces of n-BA chains with THF as the eluent, Figure S9: 1H NMR spectrum of PDMAEMA homopolymer (after purification) in CDCl3 (500 MHz, 25°C). The chemical shifts characteristic for PDMAEMA chains were assigned: δ (ppm) = 0.72-1.15 (3H, –CH3, α), 1.72-2.07 (2H, –CH2–, β), 2.23-2.53 (6H, 2 x -CH3, c), 2.57-2.84 (2H, –CH2–, b) and 3.99-4.19 (2H, –CH2–, a), Figure S10: 1H NMR spectrum of PGMA homopolymer (after purification) in CDCl3 (500 MHz, 25°C). The chemical shifts characteristic for PGMA chains were assigned: δ (ppm) = 0.80-1.18 (3H, –CH3, α), 1.84-2.06 (2H, –CH2–, β), 2.56-2.96 (2H, –CH2–, c), 3.15-3.34 (1H, –CH–, b), 3.71-3.89 and 4.19-4.39 (2H, –CH2–, a), Figure S11: 1H NMR spectrum of POEGMA homopolymer (after purification) in CDCl3 (500 MHz, 25°C). The chemical shifts characteristic for POEGMA chains were assigned: δ (ppm) = 0.78-0.97 (3H, –CH3, α), 1.23-1.30 (2H, –CH2–, β), 3.34-3.44 (3H, CH3–, c), 3.54-3.58 (2H, –CH2–, b-first segment), 3.59-3.82 (28H, –CH2– and –CH2–, a + b), and 4.01-4.17 (2H, –CH2–, a-first segment), Figure S12: 1H NMR spectrum of PBA homopolymer (after purification) in CDCl3 (500 MHz, 25°C). The chemical shifts characteristic for PBA chains were assigned: δ (ppm) = 0.85-1.01 (3H, –CH3, d), 1.31-2.00 (6H, –CH2–, a + b + c), 2.20-2.43 (1H, –CH=, α) and 3.90-4.14 (2H, –CH2–, β), Table S4: Chain-extension reaction of PDMAEMA in 7.5% coffee solution, Figure S13: Polymerization of PDMAEMA (first block) in 7.5% coffee solution (a) First-order kinetic plot of monomer conversion vs. polymerization time. (b) Mn and Mw/Mn vs. monomer conversion, Figure S14: Copolymerization of PDMAEMA (second block) in 7.5% coffee solution (a) First-order kinetic plot of monomer conversion vs. polymerization time. (b) Mn and Mw/Mn vs. monomer conversion, Figure S15: The calibration curve received for caffeine determination by DPV technique and the relative equation above the curve. The curve depicts linear dependance between the anodic peak current and caffeine concentration in 0.1 M H2SO4 aqueous solution. DP voltammograms received on GCE. DPV parameters: pulse potential of 50 mV, pulse time of 50 ms and scan rate of 50 mV/s, Figure S16: DP voltammograms received on GCE in analysis of coffee samples in 0.1 M H2SO4 aqueous solution using the standard addition method with 1 mL of the sample and after spiking 200, 400 and 600 µL of 10 mM caffeine standard solution, (a) 5% Arabica & Robusta (50/50%) sample; (b) 10% Arabica & Robusta (50/50%) sample. The analysis by standard addition method is presented in the insets. DPV parameters: pulse potential of 50 mV, pulse time of 50 ms and scan rate of 50 mV/s, Figure S17: The calibration curve received for caffeine determination by HPLC technique and the relative equation on the side. Analysis performed with the use of 1290 Infinity LC system with DAD detector and Agilent ZORBAX Eclipse Plus C18, Rapid Resolution HT, 90A, (4.6 x 50 mm, 1.8 um, 600 bar) column in 30°C. Mobile phase: water (75%) and methanol (25%), flow rate: 1.0 ml/min. Wavelength used for caffeine detection: 272 nm, Table S5. ARGET ATRP of DMAEMA in 7.0 mM caffeine solution, Figure S18: Polymerization of DMAEMA in 7.0 mM caffeine solution: (a) First-order kinetic plot of monomer conversion vs. polymerization time. (b) Mn and Mw/Mn vs. monomer conversion. (c) GPC traces of DMAEMA with DMF + 10 mM LiCl as the eluent. |
12
|
26. Karkare, P.; Kumar, S.; Murthy, C.N. ARGET-ATRP using beta-CD as reducing agent for the synthesis of PMMA-b-PS-b-PMMA triblock copolymers. J. Appl. Polym. Sci. 2019, 136 (9), 47117. 27. Chmielarz, P.; Krys, P.; Park, S.; Matyjaszewski, K. PEO-b-PNIPAM copolymers via SARA ATRP and eATRP in aqueous media. Polymer 2015, 71, 143-147. 28. Fantin, M.; Isse, A.A.; Gennaro, A.; Matyjaszewski, K. Understanding the fundamentals of aqueous ATRP and defining conditions for better control. Macromolecules 2015, 48 (19), 6862-6875. 29. Flejszar, M.; Chmielarz, P.; Smenda, J.; Wolski, K. Following principles of green chemistry: low ppm photo-ATRP of DMAEMA in water/ethanol mixture. Polymer 2021, 228, 123905. 30. Sun, Y.; Zhai, G.Q. CuSO4-catalyzed self-initiated radical polymerization of 2-(N,N-dimethylamino) ethyl methacrylate as an intrinsically reducing inimer. Chinese J. Polym. Sci. 2013, 31 (8), 1161-1172. 31. Chmielarz, P.; Krys, P.; Wang, Z.; Wang, Y.; Matyjaszewski, K. Synthesis of well-defined polymer brushes from silicon wafers via surface-initiated seATRP. Macromol. Chem. Phys. 2017, 218 (11), 1700106. 32. Morits, M.; Hynninen, V.; Nonappa; Niederberger, A.; Ikkala, O.; Gröschel, A.H.; Müllner, M. Polymer brush guided templating on well-defined rod-like cellulose nanocrystals. Polym. Chem. 2018, 9 (13), 1650-1657. 33. Xie, Y.; Chen, S.Q.; Qian, Y.H.; Zhao, W.F.; Zhao, C.S. Photo-responsive membrane surface: switching from bactericidal to bacteria-resistant property. Mater. Sci. Eng. C. 2018, 84, 52-59. 34. Xiong, D.; Zhang, X.F.; Peng, S.Y.; Gu, H.W.; Zhang, L.J. Smart pH-sensitive micelles based on redox degradable polymers as DOX/GNPs carriers for controlled drug release and CT imaging. Colloids Surf. B Biointerfaces 2018, 163, 29-40. 35. Wang, M.; Wang, X.; Zhang, K.; Wu, M.; Wu, Q.; Liu, J.; Yang, J.; Zhang, J. Nano-hydroxyapatite particle brushes via direct initiator tethering and surface-initiated atom transfer radical polymerization for dual responsive pickering emulsion. Langmuir 2020, 36 (5), 1192-1200. 36. Zhang, B.; Yan, Q.; Yuan, S.J.; Zhuang, X.D.; Zhang, F. Enhanced antifouling and anticorrosion properties of stainless steel by biomimetic anchoring PEGDMA-cross-linking polycationic brushes. Ind. Eng. Chem. Res. 2019, 58 (17), 7107-7119. 37. Chmielarz, P.; Park, S.; Simakova, A.; Matyjaszewski, K. Electrochemically mediated ATRP of acrylamides in water. Polymer 2015, 60, 302-307. 38. Michieletto, A.; Lorandi, F.; De Bon, F.; Isse, A.A.; Gennaro, A. Biocompatible polymers via aqueous electrochemically mediated atom transfer radical polymerization. J. Polym. Sci. 2020, 58 (1), 114-123. 39. Ahmed Fawzy, I.A.Z., Khalid S. Khairou, Layla S. Almazroai, Tahani M. Bawazeer, Badriah A. Al-Jahdali. Oxidation of caffeine by permanganate Ion in perchloric and sulfuric acids solutions: a comparative kinetic study. Sci. J. Chem. 2016, 4, 19-28. 40. Burton, I.W.; Farina, C.F.M.; Ragupathy, S.; Arunachalam, T.; Newmaster, S.; Berrue, F. Quantitative NMR methodology for the authentication of roasted coffee and prediction of blends. J. Agric. Food Chem. 2020, 68 (49), 14643-14651. 41. Redivo, L.; Stredansky, M.; De Angelis, E.; Navarini, L.; Resmini, M.; Svorc, L. Bare carbon electrodes as simple and efficient sensors for the quantification of caffeine in commercial beverages. R. Soc. Open Sci. 2018, 5 (5), 172146. 42. Švorc, L.u.; Tomčík, P.; Svítková, J.; Rievaj, M.; Bustin, D. Voltammetric determination of caffeine in beverage samples on bare boron-doped diamond electrode. Food Chem. 2012, 135 (3), 1198-1204. 43. Xia, J.H.; Matyjaszewski, K. Controlled/"living" radical polymerization. Atom transfer radical polymerization using multidentate amine ligands. Macromolecules 1997, 30 (25), 7697-7700. 44. Chmielarz, P.; Sobkowiak, A. Synthesis of poly(butyl acrylate) using an electrochemically mediated atom transfer radical polymerization. Polimery 2016, 61 (9), 585-590. 45. Noein, L.; Haddadi-Asl, V.; Salami-Kalajahi, M. Grafting of pH-sensitive poly (N,N-dimethylaminoethyl methacrylate-co-2-hydroxyethyl methacrylate) onto HNTS via surface-initiated atom transfer radical polymerization for controllable drug release. Int. J. Polym. Mater. Polym. Biomat. 2017, 66 (3), 123-131. 46. Zhang, M.M.; Xiong, Q.Q.; Chen, J.Q.; Wang, Y.S.; Zhang, Q.Q. A novel cyclodextrin-containing pH-responsive star polymer for nanostructure fabrication and drug delivery. Polym. Chem. 2013, 4 (19), 5086-5095. 47. Ding, A.S.; Xu, J.; Gu, G.X.; Lu, G.L.; Huang, X.Y. PHEA-g-PMMA well-defined graft copolymer: ATRP synthesis, self-assembly, and synchronous encapsulation of both hydrophobic and hydrophilic guest molecules. Sci Rep 2017, 7, 12601.
|
26. Karkare, P.; Kumar, S.; Murthy, C.N. ARGET-ATRP using beta-CD as reducing agent for the synthesis of PMMA-b-PS-b-PMMA triblock copolymers. J. Appl. Polym. Sci. 2019, 136 (9), 47117. 27. Wei, Q.; Sun, M.; Lorandi, F.; Yin, R.; Yan, J.; Liu, T.; Kowalewski, T.; Matyjaszewski, K. Cu-catalyzed atom transfer radical polymerization in the presence of liquid metal micro/nanodroplets. Macromolecules 2021, 54 (4), 1631-1638. 28. Chmielarz, P.; Krys, P.; Park, S.; Matyjaszewski, K. PEO-b-PNIPAM copolymers via SARA ATRP and eATRP in aqueous media. Polymer 2015, 71, 143-147. 29. Fantin, M.; Isse, A.A.; Gennaro, A.; Matyjaszewski, K. Understanding the fundamentals of aqueous ATRP and defining conditions for better control. Macromolecules 2015, 48 (19), 6862-6875. 30. Flejszar, M.; Chmielarz, P.; Smenda, J.; Wolski, K. Following principles of green chemistry: low ppm photo-ATRP of DMAEMA in water/ethanol mixture. Polymer 2021, 228, 123905. 31. Sun, Y.; Zhai, G.Q. CuSO4-catalyzed self-initiated radical polymerization of 2-(N,N-dimethylamino) ethyl methacrylate as an intrinsically reducing inimer. Chinese J. Polym. Sci. 2013, 31 (8), 1161-1172. 32. Chmielarz, P.; Krys, P.; Wang, Z.; Wang, Y.; Matyjaszewski, K. Synthesis of well-defined polymer brushes from silicon wafers via surface-initiated seATRP. Macromol. Chem. Phys. 2017, 218 (11), 1700106. 33. Morits, M.; Hynninen, V.; Nonappa; Niederberger, A.; Ikkala, O.; Gröschel, A.H.; Müllner, M. Polymer brush guided templating on well-defined rod-like cellulose nanocrystals. Polym. Chem. 2018, 9 (13), 1650-1657. 34. Xie, Y.; Chen, S.Q.; Qian, Y.H.; Zhao, W.F.; Zhao, C.S. Photo-responsive membrane surface: switching from bactericidal to bacteria-resistant property. Mater. Sci. Eng. C. 2018, 84, 52-59. 35. Xiong, D.; Zhang, X.F.; Peng, S.Y.; Gu, H.W.; Zhang, L.J. Smart pH-sensitive micelles based on redox degradable polymers as DOX/GNPs carriers for controlled drug release and CT imaging. Colloids Surf. B Biointerfaces 2018, 163, 29-40. 36. Wang, M.; Wang, X.; Zhang, K.; Wu, M.; Wu, Q.; Liu, J.; Yang, J.; Zhang, J. Nano-hydroxyapatite particle brushes via direct initiator tethering and surface-initiated atom transfer radical polymerization for dual responsive pickering emulsion. Langmuir 2020, 36 (5), 1192-1200. 37. Zhang, B.; Yan, Q.; Yuan, S.J.; Zhuang, X.D.; Zhang, F. Enhanced antifouling and anticorrosion properties of stainless steel by biomimetic anchoring PEGDMA-cross-linking polycationic brushes. Ind. Eng. Chem. Res. 2019, 58 (17), 7107-7119. 38. Chmielarz, P.; Park, S.; Simakova, A.; Matyjaszewski, K. Electrochemically mediated ATRP of acrylamides in water. Polymer 2015, 60, 302-307. 39. Michieletto, A.; Lorandi, F.; De Bon, F.; Isse, A.A.; Gennaro, A. Biocompatible polymers via aqueous electrochemically mediated atom transfer radical polymerization. J. Polym. Sci. 2020, 58 (1), 114-123. 40. Ahmed Fawzy, I.A.Z., Khalid S. Khairou, Layla S. Almazroai, Tahani M. Bawazeer, Badriah A. Al-Jahdali. Oxidation of caffeine by permanganate Ion in perchloric and sulfuric acids solutions: a comparative kinetic study. Sci. J. Chem. 2016, 4, 19-28. 41. Krys, P.; Matyjaszewski, K. Kinetics of atom transfer radical polymerization. Eur. Polym. J. 2017, 89, 482-523. 42. Fantin, M.; Chmielarz, P.; Wang, Y.; Lorandi, F.; Isse, A.A.; Gennaro, A.; Matyjaszewski, K. Harnessing the interaction between surfactant and hydrophilic catalyst to control eATRP in miniemulsion. Macromolecules 2017, 50 (9), 3726-3732. 43. Ribelli, T.G.; Lorandi, F.; Fantin, M.; Matyjaszewski, K. Atom transfer radical polymerization: billion times more active catalysts and new initiation systems. Macromol. Rapid Commun. 2019, 40 (1), 1800616. 44. Chmielarz, P.; Sobkowiak, A.; Matyjaszewski, K. A simplified electrochemically mediated ATRP synthesis of PEO-b-PMMA copolymers. Polymer 2015, 77, 266-271. 45. Burton, I.W.; Farina, C.F.M.; Ragupathy, S.; Arunachalam, T.; Newmaster, S.; Berrue, F. Quantitative NMR methodology for the authentication of roasted coffee and prediction of blends. J. Agric. Food Chem. 2020, 68 (49), 14643-14651. 46. Švorc, L.u.; Tomčík, P.; Svítková, J.; Rievaj, M.; Bustin, D. Voltammetric determination of caffeine in beverage samples on bare boron-doped diamond electrode. Food Chem. 2012, 135 (3), 1198-1204. 47. Redivo, L.; Stredansky, M.; De Angelis, E.; Navarini, L.; Resmini, M.; Svorc, L. Bare carbon electrodes as simple and efficient sensors for the quantification of caffeine in commercial beverages. R. Soc. Open Sci. 2018, 5 (5), 172146. 48. Xia, J.H.; Matyjaszewski, K. Controlled/"living" radical polymerization. Atom transfer radical polymerization using multidentate amine ligands. Macromolecules 1997, 30 (25), 7697-7700. 49. Chmielarz, P.; Sobkowiak, A. Synthesis of poly(butyl acrylate) using an electrochemically mediated atom transfer radical polymerization. Polimery 2016, 61 (9), 585-590. 50. Noein, L.; Haddadi-Asl, V.; Salami-Kalajahi, M. Grafting of pH-sensitive poly (N,N-dimethylaminoethyl methacrylate-co-2-hydroxyethyl methacrylate) onto HNTS via surface-initiated atom transfer radical polymerization for controllable drug release. Int. J. Polym. Mater. Polym. Biomat. 2017, 66 (3), 123-131. 51. Zhang, M.M.; Xiong, Q.Q.; Chen, J.Q.; Wang, Y.S.; Zhang, Q.Q. A novel cyclodextrin-containing pH-responsive star polymer for nanostructure fabrication and drug delivery. Polym. Chem. 2013, 4 (19), 5086-5095. 52. Ding, A.S.; Xu, J.; Gu, G.X.; Lu, G.L.; Huang, X.Y. PHEA-g-PMMA well-defined graft copolymer: ATRP synthesis, self-assembly, and synchronous encapsulation of both hydrophobic and hydrophilic guest molecules. Sci Rep 2017, 7, 12601. |
15-16 |

Reviewer 3 Report
The present paper describes the use of coffee containing beverages for the synthesis of various polymethacrylate polymers.
The authors should also consider using various tea extracts for the same reactions but in separate work/paper.
The paper is compose / written well and I would recommend to publish it as presented.
Author Response
Manuscript number: molecules-1529687
Coffee beverage: A new strategy for the synthesis of polymethacrylates via ATRP
We thank the reviewers for their valuable comments and suggestions. All changes and amendments are highlighted in the new text. Our point-by-point response follows:
Response to Reviewer 3:
Comment 1: The authors should also consider using various tea extracts for the same reactions but in separate work/paper.
Thank you for your suggestion, we will certainly take it into account when planning future experiments
Additionally, English language and style was checked carefully. All corrections have been made as follows:
Before correction |
Corrected version in revised text |
Page of the revised text (highlighted version) |
Coffee − the most popular beverage in the 21st-century society … |
Coffee − the most popular beverage in the 21st- 21st century society |
1 |
|
|
|
pure Arabica beans and proportional blend of Arabica and Robusta beans |
pure Arabica beans and a proportional blend of Arabica and Robusta beans |
1 |
…significant antioxidant capacity |
…significant antioxidant activitycapacity |
1 |
The highest amount of compounds with antioxidant potential was proved in medium roast coffee due to..… |
The highest amount of compounds with antioxidant potential was determinedproved in medium roast coffee grains due to… |
1 |
There are over five hundred types of coffee |
There are over five hundred typesspecies of coffee |
1 |
So far, mainly vitamins such as ascorbic acid [20-23] or riboflavin [12, 24] and sugars such as glucose [25] and cyclodextrin [26]. |
So far, mainly used vitamins such as ascorbic acid [20-23] or riboflavin [12, 24] and sugars such as glucose [25] and cyclodextrin [26], but also the novel external reducing agents as liquid metal [27] were used. |
2 |
Equally important is the solubility of the monomers in the individual media, which affects the choice of the reaction solvent. |
Equally important is the solubility of the monomers in the individual media, which determines the selectionaffects the choice of the reaction solvent. |
2 |
Antioxidants derived from a coffee extract as caffeine, chlorogenic acids, cafestol, kahweol and trigonelline trigger a reduction of a deactivator complex, X-CuII/L, ensuring continuous catalytic reaction. |
Antioxidants derived from a coffee extract such as caffeine, chlorogenic acids, cafestol, kahweol and trigonelline trigger a reduction of a deactivator complex, X-CuII/L, ensuring continuous catalytic reaction. Voltammetric measurements were carried out, which confirmed the strongest antioxidant effect of caffeine in both mixtures, therefore the mechanism in which caffeine is the dominant antioxidant factor controlling the polymerization process was presented. |
3 |
Caffeine, being the main antioxidant of coffee,… |
Caffeine, being the main antioxidant present inof coffee,… |
3 |
The process takes place in a mild oxidizing environment of copper (II) bromide, … |
The process takes place in a mild oxidizing environmentmedium of copper(II) bromide, … |
3 |
The mechanism involves the attack of the oxidant - deactivator on the caffeine molecule to give a free radical derived from caffeine. |
The mechanism involves the attackinteraction of the oxidant - deactivator onwith the caffeine molecule to give a free radical derived from caffeine. |
3 |
Then, water particles from the reaction solvent react with a free radical, giving 8-hydroxy caffeine which fast tautemerise to the final oxidation product of caffeine - 1,3,7-trimethyluric acid. [40] |
Then, water particles from the reaction solvent react with a free radical, giving 8-hydroxy caffeine which fast tautemerisetautomerise to the final oxidation product of caffeine - 1,3,7-trimethyluric acid (Figure 1) [40]. According to the ATRP mechanism, the dynamic equilibrium of the copper(I) and copper(II) complex is involved in controlling polymerization. The complex of copper(II) introduced into the polymerization system undergoes reduction due to the caf-feine action. Additionally, trace amounts of oxygen could cause oxidation of the copper and affect the dynamic equilibrium of CuII to CuI estabilishing. The polymerization process is then controlled by appropriate ratio of activator to deactivator species [8, 23]. |
3 |
Therefore, all next polymerization processes were performed at room temperature, allowing to achieve products in controlled manner and in a shorter time. For the synthesis in pure water without the addition of a reducing agent |
Therefore, all next polymerization processes were performed at room temperature, al-lowing to achieve products in controlled manner and in a shorter time. In efficient ATRP catalyst system, metal center must have two easily accessible oxidation states as well as must be able to accept the halogen, which requires a suffi-ciently high halogenophilicity [41]. Low halidophilicity i.e. the association of the halide an-ion to the catalyst in the higher oxidation state, related to the connection of the halo-gen ion create possibility of disproportionation of CuI/L to X-CuII/L and Cu0 [42, 43]. Finally, halide anions must have high affinity for CuIIL2+, such as to provide the pres-ence of enough concentration of deactivator to give a well-controlled process. In aqueous systems, the X-CuII/L bond can easily dissociate to CuIIL2+ and Br−, typically required is high concentration of catalyst is needed or the presence of salts with halide anions in order to shift the equilibrium toward the X-CuII/L deactivator species [28, 44]. Because of experiments were carried out in low ppm concentration of catalyst (9.5 ppm by wt%), the effect of the addition of a halide salt on the polymerization process in the medium of a coffee blend extract was investigated (Arabica & Robusta 1:1). Indeed, the lack of a halide salt caused a loss of process control due to ineffective catalyst regeneration, the addition of NaBr resulted in an increase of control over polymerization reflected valid initiation efficiency values and a narrower molecular weight distribution (Table S2). Higher salt concentration affected more effective catalyst re-generation and an increase in control over the process (compare initiation efficiency in Table S2), however, the reaction stopped after 0.5 hour did not reach a satisfactory MWD (Figure S3b), therefore it was decided to conduct subsequent reactions at a sodium bromide concentration equal to 0.1 M. For the synthesis in pure water without the addition of a reducing agent |
4 |
On the other hand, the N-CH2 group remains the initiation site of radical polymerization, resulting in a loss of ATP control as indicated by an almost 900% initiation efficiency. |
On the other hand, the N-CH2 group remains the initiation site of radical polymerization, resulting in a loss of ATPATRP control as indicated by an almost 900% initiation efficiency. |
4 |
Table 1. footer: 1 General reaction conditions: [DMAEMA]0/[EBiB]0/[CuIIBr2]0/[TPMA]0: 200/1/0.01/0.02; T = 22°C; Vtot = 8 mL (DMAEMA/solvent = 0.4/0.6 by v/v), [DMAEMA]0 = 19 mM, [I]0 = 0.09 mM. [CuIIBr2]0 = 1.19 µM (2-fold excess of TPMA), t = 0.5 h, except entry 1: t = 2h. |
Table 1. footer: 1 General reaction conditions: [DMAEMA]0/[EBiB]0/[CuIIBr2]0/[TPMA]0: 200/1/0.01/0.02; T = 22°C; Vtot = 8 mL (DMAEMA/solvent = 0.4/0.6 by v/v), [DMAEMA]0 = 19 mM, [I]0 = 0.09 mM. [NaBr] = 0.1 M, [CuIIBr2]0 = 1.19 µM (2-fold excess of TPMA), t = 0.5 h, except entry 1: t = 2h. |
5 |
caused by low amount of catalytic complex – 9.5 ppm by wt%, obtained PDMAEMA |
caused by low amount of catalytic complex – 9.5 ppm by wt%, obtained PDMAEMA |
5 |
Both types of solutions received from different blends of coffee grains were analyzed by 1H NMR (Figure S4) to confirm |
Both types of solutions received from different blends of coffee grains were analyzed by 1H NMR (Figure S4Figure S5) to confirm |
5 |
Table 2. footer: 1 General reaction conditions: [DMAEMA]0/[EBiB]0/[CuIIBr2]0/[TPMA]0: 200/1/0.01/0.02; T = 22°C; Vtot = 8 mL (DMAEMA/solvent = 0.4/0.6 by v/v), [DMAEMA]0 = 19 mM, [I]0 = 0.09 mM. [CuIIBr2]0 = 1.19 µM (2-fold excess of TPMA), t = 0.5 h. |
Table 2. footer: 1 General reaction conditions: [DMAEMA]0/[EBiB]0/[CuIIBr2]0/[TPMA]0: 200/1/0.01/0.02; T = 22°C; Vtot = 8 mL (DMAEMA/solvent = 0.4/0.6 by v/v), [DMAEMA]0 = 19 mM, [I]0 = 0.09 mM. [NaBr] = 0.1 M, [CuIIBr2]0 = 1.19 µM (2-fold excess of TPMA), t = 0.5 h. |
6 |
Table 3. footer: 1General reaction conditions: [Monomer]0/[EBiB]0: 200/1; [CuIIBr2]= 9.5 ppm by wt (2-fold excess of TPMA); T = 22°C; Vtot = 8 mL (monomer/solvent = 0.4/0.6 by v/v). |
Table 3. footer: 1General reaction conditions: [Monomer]0/[EBiB]0: 200/1; [CuIIBr2]= 9.5 ppm by wt (2-fold excess of TPMA); [NaBr] = 0.1 M, T = 22°C; Vtot = 8 mL (monomer/solvent = 0.4/0.6 by v/v). |
7 |
In order to investigate the chain-end fidelity, chain extension experiment of PDMAEMA were performed. |
In order to investigate the chain-end fidelity, chain extension experiment of PDMAEMA werewas performed. |
8 |
The PDMAEMA constituting the first block of the copolymers was characterized by monomodal GPC trace which points to the high retention of chain-end functionality (Figure 5, Figure S12). |
The PDMAEMA constituting the first block of the copolymers was characterized by monomodal GPC trace which points to the high retention of chain-end functionality (Figure 5, Figure S1213). |
8 |
The second block was successfully incorporated in situ in a controlled manner proved by linear first-order kinetics (Figure S13) and a shift in the MW peak toward to higher molecular mass (Figure 5). |
The second block was successfully incorporated in situ in a controlled manner proved by linear first-order kinetics (Figure S1314) and a shift in the MW peak toward to higher molecular mass (Figure 5). |
8 |
Three independent measurements were made for the all points included on the graphs (insets in the Figure 7 and Figure S15). |
Three independent measurements were made for the all points included on the graphs (insets in the Figure 7 and Figure S1516). |
9 |
The caffeine was successfully used as an electron donor in the ARGET ATRP technique, which also confirms reaction carried out in 7.0 mM pure caffeine solution (Table S4). |
The caffeine was successfully used as an electron donor in the ARGET ATRP technique, which also confirms reaction carried out in 7.0 mM pure caffeine solution (Table S4Table S5). |
10 |
The calibration curve equation is reported in point S.8. |
The calibration curve equation is reported in point S.8 in Supplementary Materials. |
12 |
Table S1: ARGET ATRP of DMAEMA in various temperature, Figure S1: (a) Semilogarithmic kinetic plot, (b) Mn and Mw/Mn vs. monomer conversion for preparation of PDMAEMA in coffee solution at various temperature (Table S1, entries 1-2). Reaction conditions as in Table S1, Figure S2: GPC traces of PDMAEMA chains prepared according to (a) Table S1 entry 1 and (b) Table S1 entry 2 using as eluent DMF + 10 mM LiCl, Figure S3: GPC traces of PDMAEMA chains prepared according to (a) Table 1 entry 1; (b) Table 1 entry 2 and (c) Table 1 entry 3 using as eluent DMF + 10 mM LiCl, Figure S4: 1H NMR stack plot of coffee samples: Segafredo Arabica and Segafredo Espresso CASA in D2O (500 MHz, 25°C), Figure S5: GPC traces of PDMAEMA chains prepared according to (a) Table 2 entry 1; (b) Table 2 entry 2 and (c) Table 2 entry 3 using as eluent DMF + 10 mM LiCl, Figure S6: GPC traces of (a) PDMAEMA; (b) PGMA; (c) POEGMA chains prepared according to (a) Table 3 entry 1; (b) Table 3 entry 2 and (c) Table 3 entry 3 using as eluent DMF + 10 mM LiCl, Table S2: ARGET ATRP of n-BA in miniemulsion medium of 10% coffee solution, Figure S7: Polymerization of n-BA in miniemulsion system: (a) First-order kinetic plot of monomer conversion vs. polymerization time. (b) Mn and Mw/Mn vs. monomer conversion. (c) GPC traces of n-BA chains with THF as the eluent, Figure S8: 1H NMR spectrum of PDMAEMA homopolymer (after purification) in CDCl3 (500 MHz, 25°C). The chemical shifts characteristic for PDMAEMA chains were assigned: δ (ppm) = 0.72-1.15 (3H, –CH3, α), 1.72-2.07 (2H, –CH2–, β), 2.23-2.53 (6H, 2 x -CH3, c), 2.57-2.84 (2H, –CH2–, b) and 3.99-4.19 (2H, –CH2–, a), Figure S9: 1H NMR spectrum of PGMA homopolymer (after purification) in CDCl3 (500 MHz, 25°C). The chemical shifts characteristic for PGMA chains were assigned: δ (ppm) = 0.80-1.18 (3H, –CH3, α), 1.84-2.06 (2H, –CH2–, β), 2.56-2.96 (2H, –CH2–, c), 3.15-3.34 (1H, –CH–, b), 3.71-3.89 and 4.19-4.39 (2H, –CH2–, a), Figure S10: 1H NMR spectrum of POEGMA homopolymer (after purification) in CDCl3 (500 MHz, 25°C). The chemical shifts characteristic for POEGMA chains were assigned: δ (ppm) = 0.78-0.97 (3H, –CH3, α), 1.23-1.30 (2H, –CH2–, β), 3.34-3.44 (3H, CH3–, c), 3.54-3.58 (2H, –CH2–, b-first segment), 3.59-3.82 (28H, –CH2– and –CH2–, a + b), and 4.01-4.17 (2H, –CH2–, a-first segment), Figure S11: 1H NMR spectrum of PBA homopolymer (after purification) in CDCl3 (500 MHz, 25°C). The chemical shifts characteristic for PBA chains were assigned: δ (ppm) = 0.85-1.01 (3H, –CH3, d), 1.31-2.00 (6H, –CH2–, a + b + c), 2.20-2.43 (1H, –CH=, α) and 3.90-4.14 (2H, –CH2–, β), Table S3: Chain-extension reaction of PDMAEMA in 7.5% coffee solution, Figure S12: Polymerization of PDMAEMA (first block) in 7.5% coffee solution (a) First-order kinetic plot of monomer conversion vs. polymerization time. (b) Mn and Mw/Mn vs. monomer conversion, Figure S13: Copolymerization of PDMAEMA (second block) in 7.5% coffee solution (a) First-order kinetic plot of monomer conversion vs. polymerization time. (b) Mn and Mw/Mn vs. monomer conversion, Figure S14: The calibration curve received for caffeine determination by DPV technique and the relative equation above the curve. The curve depicts linear dependance between the anodic peak current and caffeine concentration in 0.1 M H2SO4 aqueous solution. DP voltammograms received on GCE. DPV parameters: pulse potential of 50 mV, pulse time of 50 ms and scan rate of 50 mV/s, Figure S15: DP voltammograms received on GCE in analysis of coffee samples in 0.1 M H2SO4 aqueous solution using the standard addition method with 1 mL of the sample and after spiking 200, 400 and 600 µL of 10 mM caffeine standard solution, (a) 5% Arabica & Robusta (50/50%) sample; (b) 10% Arabica & Robusta (50/50%) sample. The analysis by standard addition method is presented in the insets. DPV parameters: pulse potential of 50 mV, pulse time of 50 ms and scan rate of 50 mV/s, Figure S16: The calibration curve received for caffeine determination by HPLC technique and the relative equation on the side. Analysis performed with the use of 1290 Infinity LC system with DAD detector and Agilent ZORBAX Eclipse Plus C18, Rapid Resolution HT, 90A, (4.6 x 50 mm, 1.8 um, 600 bar) column in 30°C. Mobile phase: water (75%) and methanol (25%), flow rate: 1.0 ml/min. Wavelength used for caffeine detection: 272 nm, Figure S17. Polymerization of DMAEMA in 7.0 mM caffeine solution: (a) First-order kinetic plot of monomer conversion vs. polymerization time. (b) Mn and Mw/Mn vs. monomer conversion. (c) GPC traces of DMAEMA with DMF + 10 mM LiCl as the eluent. |
Table S1: ARGET ATRP of DMAEMA in various temperature, Figure S1: (a) Semilogarithmic kinetic plot, (b) Mn and Mw/Mn vs. monomer conversion for preparation of PDMAEMA in coffee solution at various temperature (Table S1, entries 1-2). Reaction conditions as in Table S1, Figure S2: GPC traces of PDMAEMA chains prepared according to (a) Table S1 entry 1 and (b) Table S1 entry 2 using as eluent DMF + 10 mM LiCl, Table S2. ARGET ATRP in various NaBr concentration in 7.5% solution received from equally proportional blend of Arabica and Robusta beans, Figure S3. (a) Semilogarithmic kinetic plot, (b) Mn and Mw/Mn vs. monomer conversion for ARGET ATRP in preparation of PDMAEMA in various NaBr concentration (Table S2). Reactions conditions as in Table S2. Figure S3: GPC traces of PDMAEMA chains prepared according to (a) Table 1 entry 1; (b) Table 1 entry 2 and (c) Table 1 entry 3 using as eluent DMF + 10 mM LiCl, Figure S4: 1H NMR stack plot of coffee samples: Segafredo Arabica and Segafredo Espresso CASA in D2O (500 MHz, 25°C), Figure S5: GPC traces of PDMAEMA chains prepared according to (a) Table 2 entry 1; (b) Table 2 entry 2 and (c) Table 2 entry 3 using as eluent DMF + 10 mM LiCl, Figure S6: GPC traces of (a) PDMAEMA; (b) PGMA; (c) POEGMA chains prepared according to (a) Table 3 entry 1; (b) Table 3 entry 2 and (c) Table 3 entry 3 using as eluent DMF + 10 mM LiCl, Table S2: ARGET ATRP of n-BA in miniemulsion medium of 10% coffee solution, Figure S7: Polymerization of n-BA in miniemulsion system: (a) First-order kinetic plot of monomer conversion vs. polymerization time. (b) Mn and Mw/Mn vs. monomer conversion. (c) GPC traces of n-BA chains with THF as the eluent, Figure S8: 1H NMR spectrum of PDMAEMA homopolymer (after purification) in CDCl3 (500 MHz, 25°C). The chemical shifts characteristic for PDMAEMA chains were assigned: δ (ppm) = 0.72-1.15 (3H, –CH3, α), 1.72-2.07 (2H, –CH2–, β), 2.23-2.53 (6H, 2 x -CH3, c), 2.57-2.84 (2H, –CH2–, b) and 3.99-4.19 (2H, –CH2–, a), Figure S9: 1H NMR spectrum of PGMA homopolymer (after purification) in CDCl3 (500 MHz, 25°C). The chemical shifts characteristic for PGMA chains were assigned: δ (ppm) = 0.80-1.18 (3H, –CH3, α), 1.84-2.06 (2H, –CH2–, β), 2.56-2.96 (2H, –CH2–, c), 3.15-3.34 (1H, –CH–, b), 3.71-3.89 and 4.19-4.39 (2H, –CH2–, a), Figure S10: 1H NMR spectrum of POEGMA homopolymer (after purification) in CDCl3 (500 MHz, 25°C). The chemical shifts characteristic for POEGMA chains were assigned: δ (ppm) = 0.78-0.97 (3H, –CH3, α), 1.23-1.30 (2H, –CH2–, β), 3.34-3.44 (3H, CH3–, c), 3.54-3.58 (2H, –CH2–, b-first segment), 3.59-3.82 (28H, –CH2– and –CH2–, a + b), and 4.01-4.17 (2H, –CH2–, a-first segment), Figure S11: 1H NMR spectrum of PBA homopolymer (after purification) in CDCl3 (500 MHz, 25°C). The chemical shifts characteristic for PBA chains were assigned: δ (ppm) = 0.85-1.01 (3H, –CH3, d), 1.31-2.00 (6H, –CH2–, a + b + c), 2.20-2.43 (1H, –CH=, α) and 3.90-4.14 (2H, –CH2–, β), Table S3: Chain-extension reaction of PDMAEMA in 7.5% coffee solution, Figure S12: Polymerization of PDMAEMA (first block) in 7.5% coffee solution (a) First-order kinetic plot of monomer conversion vs. polymerization time. (b) Mn and Mw/Mn vs. monomer conversion, Figure S13: Copolymerization of PDMAEMA (second block) in 7.5% coffee solution (a) First-order kinetic plot of monomer conversion vs. polymerization time. (b) Mn and Mw/Mn vs. monomer conversion, Figure S14: The calibration curve received for caffeine determination by DPV technique and the relative equation above the curve. The curve depicts linear dependance between the anodic peak current and caffeine concentration in 0.1 M H2SO4 aqueous solution. DP voltammograms received on GCE. DPV parameters: pulse potential of 50 mV, pulse time of 50 ms and scan rate of 50 mV/s, Figure S15: DP voltammograms received on GCE in analysis of coffee samples in 0.1 M H2SO4 aqueous solution using the standard addition method with 1 mL of the sample and after spiking 200, 400 and 600 µL of 10 mM caffeine standard solution, (a) 5% Arabica & Robusta (50/50%) sample; (b) 10% Arabica & Robusta (50/50%) sample. The analysis by standard addition method is presented in the insets. DPV parameters: pulse potential of 50 mV, pulse time of 50 ms and scan rate of 50 mV/s, Figure S16: The calibration curve received for caffeine determination by HPLC technique and the relative equation on the side. Analysis performed with the use of 1290 Infinity LC system with DAD detector and Agilent ZORBAX Eclipse Plus C18, Rapid Resolution HT, 90A, (4.6 x 50 mm, 1.8 um, 600 bar) column in 30°C. Mobile phase: water (75%) and methanol (25%), flow rate: 1.0 ml/min. Wavelength used for caffeine detection: 272 nm, Figure S17. Polymerization of DMAEMA in 7.0 mM caffeine solution: (a) First-order kinetic plot of monomer conversion vs. polymerization time. (b) Mn and Mw/Mn vs. monomer conversion. (c) GPC traces of DMAEMA with DMF + 10 mM LiCl as the eluent. Figure S4: GPC traces of PDMAEMA chains prepared according to (a) Table 1 entry 1; (b) Table 1 entry 2 and (c) Table 1 entry 3 using as eluent DMF + 10 mM LiCl, Figure S5: 1H NMR stack plot of coffee samples: Segafredo Arabica and Segafredo Espresso CASA in D2O (500 MHz, 25°C), Figure S6: GPC traces of PDMAEMA chains prepared according to (a) Table 2 entry 1; (b) Table 2 entry 2 and (c) Table 2 entry 3 using as eluent DMF + 10 mM LiCl, Figure S7: GPC traces of (a) PDMAEMA; (b) PGMA; (c) POEGMA chains prepared according to (a) Table 3 entry 1; (b) Table 3 entry 2 and (c) Table 3 entry 3 using as eluent DMF + 10 mM LiCl, Table S3: ARGET ATRP of n-BA in miniemulsion medium of 10% coffee solution, Figure S8: Polymerization of n-BA in miniemulsion system: (a) First-order kinetic plot of monomer conversion vs. polymerization time. (b) Mn and Mw/Mn vs. monomer conversion. (c) GPC traces of n-BA chains with THF as the eluent, Figure S9: 1H NMR spectrum of PDMAEMA homopolymer (after purification) in CDCl3 (500 MHz, 25°C). The chemical shifts characteristic for PDMAEMA chains were assigned: δ (ppm) = 0.72-1.15 (3H, –CH3, α), 1.72-2.07 (2H, –CH2–, β), 2.23-2.53 (6H, 2 x -CH3, c), 2.57-2.84 (2H, –CH2–, b) and 3.99-4.19 (2H, –CH2–, a), Figure S10: 1H NMR spectrum of PGMA homopolymer (after purification) in CDCl3 (500 MHz, 25°C). The chemical shifts characteristic for PGMA chains were assigned: δ (ppm) = 0.80-1.18 (3H, –CH3, α), 1.84-2.06 (2H, –CH2–, β), 2.56-2.96 (2H, –CH2–, c), 3.15-3.34 (1H, –CH–, b), 3.71-3.89 and 4.19-4.39 (2H, –CH2–, a), Figure S11: 1H NMR spectrum of POEGMA homopolymer (after purification) in CDCl3 (500 MHz, 25°C). The chemical shifts characteristic for POEGMA chains were assigned: δ (ppm) = 0.78-0.97 (3H, –CH3, α), 1.23-1.30 (2H, –CH2–, β), 3.34-3.44 (3H, CH3–, c), 3.54-3.58 (2H, –CH2–, b-first segment), 3.59-3.82 (28H, –CH2– and –CH2–, a + b), and 4.01-4.17 (2H, –CH2–, a-first segment), Figure S12: 1H NMR spectrum of PBA homopolymer (after purification) in CDCl3 (500 MHz, 25°C). The chemical shifts characteristic for PBA chains were assigned: δ (ppm) = 0.85-1.01 (3H, –CH3, d), 1.31-2.00 (6H, –CH2–, a + b + c), 2.20-2.43 (1H, –CH=, α) and 3.90-4.14 (2H, –CH2–, β), Table S4: Chain-extension reaction of PDMAEMA in 7.5% coffee solution, Figure S13: Polymerization of PDMAEMA (first block) in 7.5% coffee solution (a) First-order kinetic plot of monomer conversion vs. polymerization time. (b) Mn and Mw/Mn vs. monomer conversion, Figure S14: Copolymerization of PDMAEMA (second block) in 7.5% coffee solution (a) First-order kinetic plot of monomer conversion vs. polymerization time. (b) Mn and Mw/Mn vs. monomer conversion, Figure S15: The calibration curve received for caffeine determination by DPV technique and the relative equation above the curve. The curve depicts linear dependance between the anodic peak current and caffeine concentration in 0.1 M H2SO4 aqueous solution. DP voltammograms received on GCE. DPV parameters: pulse potential of 50 mV, pulse time of 50 ms and scan rate of 50 mV/s, Figure S16: DP voltammograms received on GCE in analysis of coffee samples in 0.1 M H2SO4 aqueous solution using the standard addition method with 1 mL of the sample and after spiking 200, 400 and 600 µL of 10 mM caffeine standard solution, (a) 5% Arabica & Robusta (50/50%) sample; (b) 10% Arabica & Robusta (50/50%) sample. The analysis by standard addition method is presented in the insets. DPV parameters: pulse potential of 50 mV, pulse time of 50 ms and scan rate of 50 mV/s, Figure S17: The calibration curve received for caffeine determination by HPLC technique and the relative equation on the side. Analysis performed with the use of 1290 Infinity LC system with DAD detector and Agilent ZORBAX Eclipse Plus C18, Rapid Resolution HT, 90A, (4.6 x 50 mm, 1.8 um, 600 bar) column in 30°C. Mobile phase: water (75%) and methanol (25%), flow rate: 1.0 ml/min. Wavelength used for caffeine detection: 272 nm, Table S5. ARGET ATRP of DMAEMA in 7.0 mM caffeine solution, Figure S18: Polymerization of DMAEMA in 7.0 mM caffeine solution: (a) First-order kinetic plot of monomer conversion vs. polymerization time. (b) Mn and Mw/Mn vs. monomer conversion. (c) GPC traces of DMAEMA with DMF + 10 mM LiCl as the eluent. |
12
|
26. Karkare, P.; Kumar, S.; Murthy, C.N. ARGET-ATRP using beta-CD as reducing agent for the synthesis of PMMA-b-PS-b-PMMA triblock copolymers. J. Appl. Polym. Sci. 2019, 136 (9), 47117. 27. Chmielarz, P.; Krys, P.; Park, S.; Matyjaszewski, K. PEO-b-PNIPAM copolymers via SARA ATRP and eATRP in aqueous media. Polymer 2015, 71, 143-147. 28. Fantin, M.; Isse, A.A.; Gennaro, A.; Matyjaszewski, K. Understanding the fundamentals of aqueous ATRP and defining conditions for better control. Macromolecules 2015, 48 (19), 6862-6875. 29. Flejszar, M.; Chmielarz, P.; Smenda, J.; Wolski, K. Following principles of green chemistry: low ppm photo-ATRP of DMAEMA in water/ethanol mixture. Polymer 2021, 228, 123905. 30. Sun, Y.; Zhai, G.Q. CuSO4-catalyzed self-initiated radical polymerization of 2-(N,N-dimethylamino) ethyl methacrylate as an intrinsically reducing inimer. Chinese J. Polym. Sci. 2013, 31 (8), 1161-1172. 31. Chmielarz, P.; Krys, P.; Wang, Z.; Wang, Y.; Matyjaszewski, K. Synthesis of well-defined polymer brushes from silicon wafers via surface-initiated seATRP. Macromol. Chem. Phys. 2017, 218 (11), 1700106. 32. Morits, M.; Hynninen, V.; Nonappa; Niederberger, A.; Ikkala, O.; Gröschel, A.H.; Müllner, M. Polymer brush guided templating on well-defined rod-like cellulose nanocrystals. Polym. Chem. 2018, 9 (13), 1650-1657. 33. Xie, Y.; Chen, S.Q.; Qian, Y.H.; Zhao, W.F.; Zhao, C.S. Photo-responsive membrane surface: switching from bactericidal to bacteria-resistant property. Mater. Sci. Eng. C. 2018, 84, 52-59. 34. Xiong, D.; Zhang, X.F.; Peng, S.Y.; Gu, H.W.; Zhang, L.J. Smart pH-sensitive micelles based on redox degradable polymers as DOX/GNPs carriers for controlled drug release and CT imaging. Colloids Surf. B Biointerfaces 2018, 163, 29-40. 35. Wang, M.; Wang, X.; Zhang, K.; Wu, M.; Wu, Q.; Liu, J.; Yang, J.; Zhang, J. Nano-hydroxyapatite particle brushes via direct initiator tethering and surface-initiated atom transfer radical polymerization for dual responsive pickering emulsion. Langmuir 2020, 36 (5), 1192-1200. 36. Zhang, B.; Yan, Q.; Yuan, S.J.; Zhuang, X.D.; Zhang, F. Enhanced antifouling and anticorrosion properties of stainless steel by biomimetic anchoring PEGDMA-cross-linking polycationic brushes. Ind. Eng. Chem. Res. 2019, 58 (17), 7107-7119. 37. Chmielarz, P.; Park, S.; Simakova, A.; Matyjaszewski, K. Electrochemically mediated ATRP of acrylamides in water. Polymer 2015, 60, 302-307. 38. Michieletto, A.; Lorandi, F.; De Bon, F.; Isse, A.A.; Gennaro, A. Biocompatible polymers via aqueous electrochemically mediated atom transfer radical polymerization. J. Polym. Sci. 2020, 58 (1), 114-123. 39. Ahmed Fawzy, I.A.Z., Khalid S. Khairou, Layla S. Almazroai, Tahani M. Bawazeer, Badriah A. Al-Jahdali. Oxidation of caffeine by permanganate Ion in perchloric and sulfuric acids solutions: a comparative kinetic study. Sci. J. Chem. 2016, 4, 19-28. 40. Burton, I.W.; Farina, C.F.M.; Ragupathy, S.; Arunachalam, T.; Newmaster, S.; Berrue, F. Quantitative NMR methodology for the authentication of roasted coffee and prediction of blends. J. Agric. Food Chem. 2020, 68 (49), 14643-14651. 41. Redivo, L.; Stredansky, M.; De Angelis, E.; Navarini, L.; Resmini, M.; Svorc, L. Bare carbon electrodes as simple and efficient sensors for the quantification of caffeine in commercial beverages. R. Soc. Open Sci. 2018, 5 (5), 172146. 42. Švorc, L.u.; Tomčík, P.; Svítková, J.; Rievaj, M.; Bustin, D. Voltammetric determination of caffeine in beverage samples on bare boron-doped diamond electrode. Food Chem. 2012, 135 (3), 1198-1204. 43. Xia, J.H.; Matyjaszewski, K. Controlled/"living" radical polymerization. Atom transfer radical polymerization using multidentate amine ligands. Macromolecules 1997, 30 (25), 7697-7700. 44. Chmielarz, P.; Sobkowiak, A. Synthesis of poly(butyl acrylate) using an electrochemically mediated atom transfer radical polymerization. Polimery 2016, 61 (9), 585-590. 45. Noein, L.; Haddadi-Asl, V.; Salami-Kalajahi, M. Grafting of pH-sensitive poly (N,N-dimethylaminoethyl methacrylate-co-2-hydroxyethyl methacrylate) onto HNTS via surface-initiated atom transfer radical polymerization for controllable drug release. Int. J. Polym. Mater. Polym. Biomat. 2017, 66 (3), 123-131. 46. Zhang, M.M.; Xiong, Q.Q.; Chen, J.Q.; Wang, Y.S.; Zhang, Q.Q. A novel cyclodextrin-containing pH-responsive star polymer for nanostructure fabrication and drug delivery. Polym. Chem. 2013, 4 (19), 5086-5095. 47. Ding, A.S.; Xu, J.; Gu, G.X.; Lu, G.L.; Huang, X.Y. PHEA-g-PMMA well-defined graft copolymer: ATRP synthesis, self-assembly, and synchronous encapsulation of both hydrophobic and hydrophilic guest molecules. Sci Rep 2017, 7, 12601.
|
26. Karkare, P.; Kumar, S.; Murthy, C.N. ARGET-ATRP using beta-CD as reducing agent for the synthesis of PMMA-b-PS-b-PMMA triblock copolymers. J. Appl. Polym. Sci. 2019, 136 (9), 47117. 27. Wei, Q.; Sun, M.; Lorandi, F.; Yin, R.; Yan, J.; Liu, T.; Kowalewski, T.; Matyjaszewski, K. Cu-catalyzed atom transfer radical polymerization in the presence of liquid metal micro/nanodroplets. Macromolecules 2021, 54 (4), 1631-1638. 28. Chmielarz, P.; Krys, P.; Park, S.; Matyjaszewski, K. PEO-b-PNIPAM copolymers via SARA ATRP and eATRP in aqueous media. Polymer 2015, 71, 143-147. 29. Fantin, M.; Isse, A.A.; Gennaro, A.; Matyjaszewski, K. Understanding the fundamentals of aqueous ATRP and defining conditions for better control. Macromolecules 2015, 48 (19), 6862-6875. 30. Flejszar, M.; Chmielarz, P.; Smenda, J.; Wolski, K. Following principles of green chemistry: low ppm photo-ATRP of DMAEMA in water/ethanol mixture. Polymer 2021, 228, 123905. 31. Sun, Y.; Zhai, G.Q. CuSO4-catalyzed self-initiated radical polymerization of 2-(N,N-dimethylamino) ethyl methacrylate as an intrinsically reducing inimer. Chinese J. Polym. Sci. 2013, 31 (8), 1161-1172. 32. Chmielarz, P.; Krys, P.; Wang, Z.; Wang, Y.; Matyjaszewski, K. Synthesis of well-defined polymer brushes from silicon wafers via surface-initiated seATRP. Macromol. Chem. Phys. 2017, 218 (11), 1700106. 33. Morits, M.; Hynninen, V.; Nonappa; Niederberger, A.; Ikkala, O.; Gröschel, A.H.; Müllner, M. Polymer brush guided templating on well-defined rod-like cellulose nanocrystals. Polym. Chem. 2018, 9 (13), 1650-1657. 34. Xie, Y.; Chen, S.Q.; Qian, Y.H.; Zhao, W.F.; Zhao, C.S. Photo-responsive membrane surface: switching from bactericidal to bacteria-resistant property. Mater. Sci. Eng. C. 2018, 84, 52-59. 35. Xiong, D.; Zhang, X.F.; Peng, S.Y.; Gu, H.W.; Zhang, L.J. Smart pH-sensitive micelles based on redox degradable polymers as DOX/GNPs carriers for controlled drug release and CT imaging. Colloids Surf. B Biointerfaces 2018, 163, 29-40. 36. Wang, M.; Wang, X.; Zhang, K.; Wu, M.; Wu, Q.; Liu, J.; Yang, J.; Zhang, J. Nano-hydroxyapatite particle brushes via direct initiator tethering and surface-initiated atom transfer radical polymerization for dual responsive pickering emulsion. Langmuir 2020, 36 (5), 1192-1200. 37. Zhang, B.; Yan, Q.; Yuan, S.J.; Zhuang, X.D.; Zhang, F. Enhanced antifouling and anticorrosion properties of stainless steel by biomimetic anchoring PEGDMA-cross-linking polycationic brushes. Ind. Eng. Chem. Res. 2019, 58 (17), 7107-7119. 38. Chmielarz, P.; Park, S.; Simakova, A.; Matyjaszewski, K. Electrochemically mediated ATRP of acrylamides in water. Polymer 2015, 60, 302-307. 39. Michieletto, A.; Lorandi, F.; De Bon, F.; Isse, A.A.; Gennaro, A. Biocompatible polymers via aqueous electrochemically mediated atom transfer radical polymerization. J. Polym. Sci. 2020, 58 (1), 114-123. 40. Ahmed Fawzy, I.A.Z., Khalid S. Khairou, Layla S. Almazroai, Tahani M. Bawazeer, Badriah A. Al-Jahdali. Oxidation of caffeine by permanganate Ion in perchloric and sulfuric acids solutions: a comparative kinetic study. Sci. J. Chem. 2016, 4, 19-28. 41. Krys, P.; Matyjaszewski, K. Kinetics of atom transfer radical polymerization. Eur. Polym. J. 2017, 89, 482-523. 42. Fantin, M.; Chmielarz, P.; Wang, Y.; Lorandi, F.; Isse, A.A.; Gennaro, A.; Matyjaszewski, K. Harnessing the interaction between surfactant and hydrophilic catalyst to control eATRP in miniemulsion. Macromolecules 2017, 50 (9), 3726-3732. 43. Ribelli, T.G.; Lorandi, F.; Fantin, M.; Matyjaszewski, K. Atom transfer radical polymerization: billion times more active catalysts and new initiation systems. Macromol. Rapid Commun. 2019, 40 (1), 1800616. 44. Chmielarz, P.; Sobkowiak, A.; Matyjaszewski, K. A simplified electrochemically mediated ATRP synthesis of PEO-b-PMMA copolymers. Polymer 2015, 77, 266-271. 45. Burton, I.W.; Farina, C.F.M.; Ragupathy, S.; Arunachalam, T.; Newmaster, S.; Berrue, F. Quantitative NMR methodology for the authentication of roasted coffee and prediction of blends. J. Agric. Food Chem. 2020, 68 (49), 14643-14651. 46. Švorc, L.u.; Tomčík, P.; Svítková, J.; Rievaj, M.; Bustin, D. Voltammetric determination of caffeine in beverage samples on bare boron-doped diamond electrode. Food Chem. 2012, 135 (3), 1198-1204. 47. Redivo, L.; Stredansky, M.; De Angelis, E.; Navarini, L.; Resmini, M.; Svorc, L. Bare carbon electrodes as simple and efficient sensors for the quantification of caffeine in commercial beverages. R. Soc. Open Sci. 2018, 5 (5), 172146. 48. Xia, J.H.; Matyjaszewski, K. Controlled/"living" radical polymerization. Atom transfer radical polymerization using multidentate amine ligands. Macromolecules 1997, 30 (25), 7697-7700. 49. Chmielarz, P.; Sobkowiak, A. Synthesis of poly(butyl acrylate) using an electrochemically mediated atom transfer radical polymerization. Polimery 2016, 61 (9), 585-590. 50. Noein, L.; Haddadi-Asl, V.; Salami-Kalajahi, M. Grafting of pH-sensitive poly (N,N-dimethylaminoethyl methacrylate-co-2-hydroxyethyl methacrylate) onto HNTS via surface-initiated atom transfer radical polymerization for controllable drug release. Int. J. Polym. Mater. Polym. Biomat. 2017, 66 (3), 123-131. 51. Zhang, M.M.; Xiong, Q.Q.; Chen, J.Q.; Wang, Y.S.; Zhang, Q.Q. A novel cyclodextrin-containing pH-responsive star polymer for nanostructure fabrication and drug delivery. Polym. Chem. 2013, 4 (19), 5086-5095. 52. Ding, A.S.; Xu, J.; Gu, G.X.; Lu, G.L.; Huang, X.Y. PHEA-g-PMMA well-defined graft copolymer: ATRP synthesis, self-assembly, and synchronous encapsulation of both hydrophobic and hydrophilic guest molecules. Sci Rep 2017, 7, 12601. |
15-16 |
